

# Early Pliocene vegetation and hydrology changes in western equatorial South America

Friederike Grimmer[1], Lydie Dupont[1], Frank Lamy[2], Gerlinde Jung[1], Catalina González[3], Gerold Wefer[1]

[1]MARUM – Center for Marine Environmental Sciences, University of Bremen, Leobener Str. 8, 28359 Bremen, Germany
[2]Alfred-Wegener-Institute for Polar and Marine Research, Am Handelshafen 12, 27570 Bremerhaven, Germany
[3]Department of Biological Sciences, Universidad de los Andes, Cra. 1 #18a-12, Bogotá, Colombia

*Correspondence to*: Friederike Grimmer (fgrimmer@marum.de)

**Abstract.** During the early Pliocene, two major tectonic events triggered a profound reorganization of ocean and atmospheric circulation in the Eastern Equatorial Pacific (EEP), the Caribbean Sea, and on adjacent land masses: the progressive closure of the Central American Seaway (CAS) and the uplift of the northern Andes. These affected amongst others the mean latitudinal position of the Intertropical Convergence Zone (ITCZ). The direction of an ITCZ shift however is still debated, as numeric modelling results and paleoceanographic data indicate shifts in opposite directions. To provide new insights into this debate, an independent hydrological record of western equatorial South America was generated. Vegetation and climate of this area were reconstructed by pollen analysis of 46 samples from marine sediment core ODP 1239A from the EEP comprising the interval between 4.7 and 4.2 Ma. The study site is sensitive to latitudinal ITCZ shifts insofar as a southward (northward) shift would result in increased (decreased) precipitation over Ecuador. The presented pollen record comprises representatives from five ecological groups: lowland rainforest, lower montane forest, upper montane forest, páramo, and broad range taxa. A broad tropical rainforest coverage persisted in the study area throughout the early Pliocene, without significant open vegetation below the forest line. Between 4.7 and 4.42 Ma, humidity increases, reaching its peak around 4.42 Ma, and slightly decreasing again afterwards. The stable, permanently humid conditions are rather in agreement with paleoceanographic data indicating a southward shift of the ITCZ, possibly in response to CAS closure. The presence of páramo vegetation indicates that the Western Cordillera of the northern Andes had already reached considerable elevation by the early Pliocene. Future studies could extend the hydrological record of the region further back into the late Miocene to see if a more profound atmospheric response to tectonic changes occurred earlier.

## 1 Introduction

The Pliocene epoch is characterized by some profound tectonic processes which altered oceanic and atmospheric circulation on a regional and possibly also global scale (Cannariato and Ravelo, 1997; Lunt et al., 2008). Two of these processes are the closure of the Isthmus of Panama and the uplift of the northern Andes. The formation of the Isthmus of Panama, and especially the precise temporal constraints of the closure of the Panama Strait, have been subject of numerous studies (Bartoli et al., 2005; Groeneveld et al., 2014; Hoorn and Flantua, 2015; Montes, 2015; Steph, 2005). A recent review based on geological,



paleontological, and molecular records narrowed the formation *sensu stricto* down to 2.8 Ma (O'Dea et al., 2016). Temporal
constraints on the restriction of the surface water flow through the gateway were established by salinity reconstructions on
both sides of the Isthmus (Steph et al., 2006b, Fig. 1). The salinities first start to diverge around 4.5 Ma. A major step in the
seaway closure between 4.7 and 4.2 Ma was also assumed based on the comparison of mass accumulation rates of the carbonate
sand-fraction in the Caribbean Sea and the EEP (Haug and Tiedemann, 1998). The closure of the Central American Seaway
has been associated with the development of the EEP cold tongue (EEP CT), strengthened upwelling in the EEP, the shoaling
of the thermocline, and a mean latitudinal shift of the ITCZ (Steph, 2005; Steph et al., 2006a; Steph et al., 2006b; Steph et al.,
2010). The direction of a potential shift of the ITCZ is still debated because of a discrepancy between paleoclimate
reconstructions based on proxy data and numerical modelling results.
For the late Miocene, a northernmost paleoposition of the ITCZ at about 10–12°N has been proposed (Flohn, 1981; Hovan,
1995). Subsequently, a southward shift towards 5°N paleolatitude between 5 and 4 Ma is indicated by eolian grain-size
distributions in the eastern tropical Pacific (Hovan, 1995). Billups et al. (1999) provide additional evidence for a southward
shift of the ITCZ between 4.4 and 4.3 Ma. Hence, most proxy data agree about a southward ITCZ shift during the early
Pliocene. On the contrary, results from numerical modelling suggest a northward shift of the ITCZ in response to CAS closure
(Steph et al., 2006b) and Andean uplift (Feng and Poulsen, 2014; Takahashi and Battisti, 2007).
An independent record of the terrestrial hydrology for the early Pliocene from a study site that is sensitive to latitudinal ITCZ
shifts could provide new insights to this debate. Schneider et al. (2014) also stress the need of reconstructions of the ITCZ in
the early and mid-Pliocene in order to understand how competing effects like an ice-free northern hemisphere and a weak EEP
CT balanced, and to reduce uncertainties of predictions. Even though changes of ocean–atmosphere linkages related to ENSO
(El Niño Southern Oscillation) and ITCZ shifts strongly impact continental precipitation in western equatorial South America,
most studies so far have focused on paleoceanographic features such as sea surface temperatures and ocean stratification.



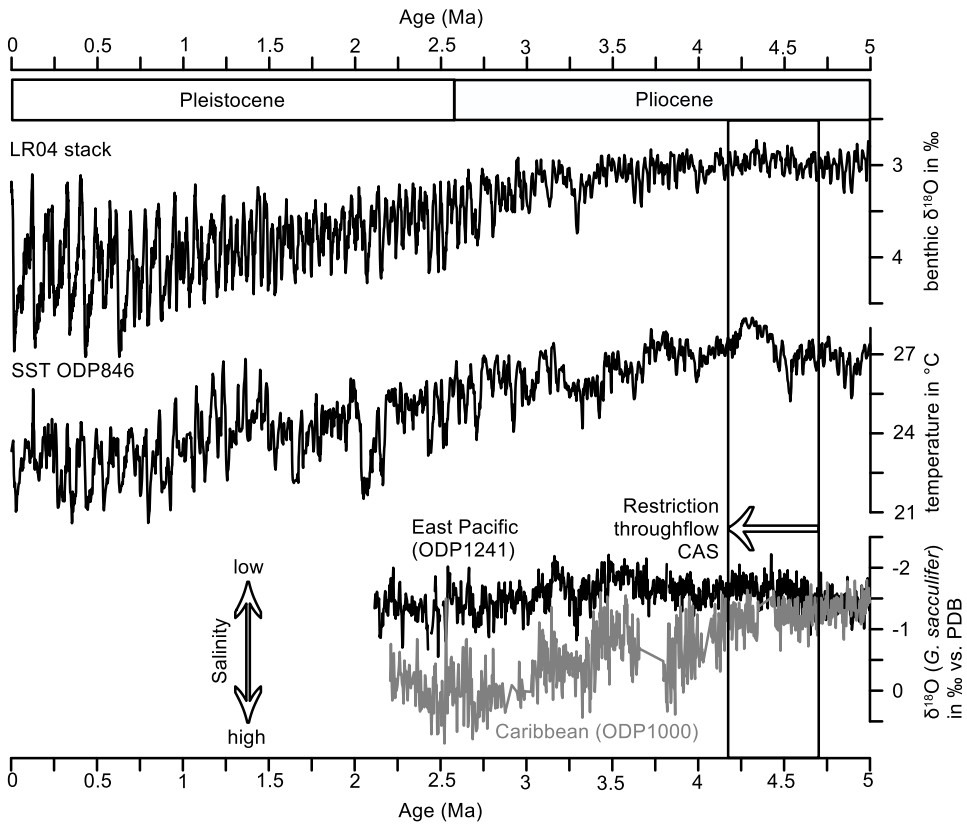


**Figure 1: LR04 global stack of benthic δ¹⁸O reflecting changes in global ice volume and temperature. Uk'37 sea surface temperatures**
**(SST) of ODP Site 846 in the Equatorial Pacific Cold Tongue (Lawrence et al., 2006). δ¹⁸O of the planktonic foraminifer *G. sacculifer***
**from ODP Site 1000 in the Caribbean and ODP Site 1241 in the East Pacific (Steph, 2005; Steph et al., 2006a), reflecting changes in**
**sea surface salinity (see Fig. 2 for location of ODP Sites). The box represents the time window analyzed in this study.**

The second major tectonic process is the uplift of the Northern Andes which strongly altered atmospheric circulation patterns

over South America. Three major deformation phases include fan building in the lower Eocene to early Oligocene, compression

of Oligocene deposits in the Miocene and Pliocene, and refolding during Pliocene to recent times (Corredor, 2003). While the

uplift of the Central Andes is well investigated, only few studies deal with the timing of uplift of the Northern Andes. Coltorti

and Ollier (2000), based on geomorphologic data, conclude that the uplift of the Ecuadorian Andes started in the early Pliocene

and continued until the Pleistocene. More recent apatite fission track data indicate that the western Andean Cordillera of

Ecuador was rapidly exhumed during the late Miocene (13–9 Ma) (Spikings et al., 2005). Uplift estimates for the Central

Andes suggest that the Altiplano had reached less than half of its modern elevation by 10 Ma, with uplift rates increasing from

0.1 mm/yr in the early and middle Miocene to 0.2–0.3 mm/yr to present. For the Eastern Cordillera of Colombia, elevations

of less than 40% of the modern values are estimated for the early Pliocene, then increasing rapidly at rates of 0.5–3 mm/yr

until modern elevations were reached around 2.7 Ma (Gregory-Wodzicki, 2000). Both the tectonic events and the closure of

the Central American Seaway are assumed to have had a large impact on ocean and atmospheric circulation in the eastern





Pacific, the Caribbean and on adjacent land masses. Therefore, the reconstruction of continental climate, especially hydrology,
will contribute to our understanding of climatic changes in this highly complex area.
To better understand the early Pliocene vegetation and hydrology of western equatorial South America we studied pollen and
spores from the early Pliocene section (4.7–4.2 Ma) of the marine sediment record at ODP Site 1239 and compared this record
to Holocene samples from the same Site. While other palynological studies of the region have been conducted for the mid-
Pliocene to Holocene (González et al., 2006; Hooghiemstra, 1984; Seilles et al., 2016), this is the first palynological record of
western equatorial South America from the early Pliocene. The record contributes to elucidate how vegetation and climate in
this area responded to changes in atmospheric and oceanic circulation, possibly induced by the closure of the Central American
Seaway and the uplift of the northern Andes. Therefore the main objectives of the study are firstly, to investigate long-term
vegetation and climatic changes, focusing on hydrology, in western equatorial South America and, secondly, to interpret these
changes in relation to climate phenomena influencing the hydrology of the region, especially the mean latitudinal position of
the ITCZ and variability related to ENSO. These objectives are approached by the following research questions: 1) What floral
and vegetation changes took place in the coastal plain and the Western Andean Cordillera of western equatorial South America
from 4.7 to 4.2 Ma? 2) What are the climatic implications of the vegetation change, especially in terms of hydrology? 3) What
are the implications for Andean uplift, especially regarding the development of the high Andean páramo vegetation?





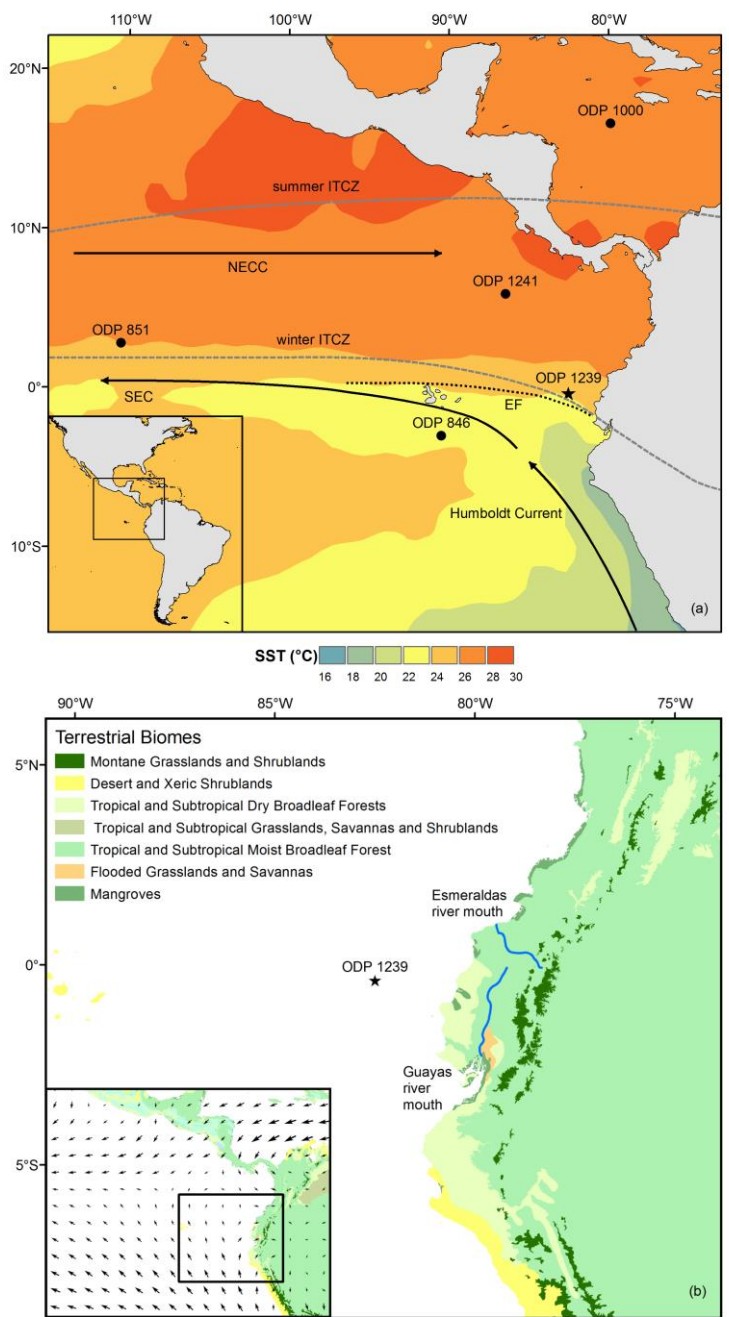

**Figure 2: (a) Major oceanographic features of the eastern equatorial Pacific (SST: Sea surface temperature, statistical annual mean from 2005-2012 from NOAA; NECC: North Equatorial Counter Current, SEC: South Equatorial Current, EF: Equatorial Front), and boreal summer and winter position of the Intertropical Convergence Zone (ITCZ). The locations of ODP Sites mentioned in the text are indicated. (b) Modern vegetation of western equatorial South America as defined by the World Wildlife Fund (please note that the terrestrial biomes are not identical to the altitudinal vegetation belts shown in Fig. 3 and 4), major rivers draining into the Pacific, and magnitude and direction of January surface winds (NCEP Reanalysis Derived monthly long term means from 1981-2010 provided by the NOAA/OAR/ESRL PSD, Boulder, Colorado, USA, from their website at http://www.esrl.noaa.gov/psd/).**



### 1.1 Modern setting

#### 1.1.1 Climate and ocean circulation

The climate of western equatorial South America is complex and heterogeneous, as it is not only controlled by large-scale tropical climate phenomena such as the ITCZ and the El Niño Southern Oscillation (ENSO), but is also strongly influenced by small-scale climate patterns caused by the diverse Andean topography. The annual cycle of precipitation in northwestern South America is controlled by insolation changes. During boreal summer when insolation is strongest in the northern hemisphere, the ITCZ is located at its northernmost position around 9°–10° N (Vuille et al., 2000). Approaching austral summer, the ITCZ moves southward across the equator. Within the range of the ITCZ, annual precipitation patterns are generally characterized by two minima and two maxima. The coastal areas of southern Ecuador where the ITCZ has its southernmost excursion show an annual precipitation pattern with one maximum during austral summer and a pronounced dry season during austral winter (Bendix and Lauer, 1992).

This general circulation pattern is modified by ENSO at interannual time-scales. During warm El Niño events, the lowlands of Ecuador experience abundant precipitation whereas the northwestern Ecuadorian Andes experience drought (Vuille et al., 2000). Regional climate patterns are also modified by the topography of the Andes which pose an effective barrier for the large-scale atmospheric circulation. While precipitation patterns east of the Andes are driven by moisture-laden easterly trade winds originating over the tropical Atlantic and the Amazon basin, the coastal areas and the western Andean slopes are dominated by air masses originating in the Pacific (Vuille et al., 2000, Fig. 2). The warm annual El Niño current which flows southward along the Colombian Pacific coast warms the air masses along the coast. This moist air brings over 6000 mm yearly precipitation to the northern coastal plain (Balslev, 1988). In contrast, the coastal areas of southernmost Ecuador and northern Peru are under the influence of the cold Humboldt Current which transports cold and nutrient rich waters and gives rise to a long strip of coastal desert. The westwards flow of the cold surface waters of the EEP CT to the western Pacific via the South Equatorial Current (SEC) is driven by the Walker Circulation. Warm waters return eastwards via the North Equatorial Countercurrent (NECC, see Fig. 2). An abrupt transition between the cold SEC and the warm NECC is the Equatorial Front (EF).

#### 1.1.2 Geography, vegetation and pollen transport

The study area is geographically divided into three main regions: the coastal plain with several rivers draining into the Pacific, the Andes, and the eastern lowlands which constitute the western margin of the Amazon Basin. The mountains form two parallel cordilleras which are separated by the Interandean Valley. The diverse vegetation is the result of the combined effects of elevation and precipitation. In the coastal plain there is an abrupt shift from tropical lowland rainforests in the north to a desert dominated by annual xerophytic herbs in the south. This shift reflects the dependence of the vegetation on precipitation which ranges from 100 to 6000 mm per year on the coastal plain. The western slopes of the Andes are covered by montane forest, which is partly interrupted by drier valleys in southern Ecuador (Balslev, 1988).





Along the coast, mangrove stands occur in the salt- and brackish-water tidal zone of river estuaries and bays. They are formed
by two species of *Rhizophora* (*R. harrisonii* and *R. mangle*), and to a lesser extent *Avicennia, Laguncularia*, and *Conocarpus*
are present (Twilley et al., 2001). The lowland rainforest is characterized by the dominant plant families Fabaceae, Rubiaceae,
Palmae, Annonaceae, Melastomataceae, Sapotaceae, and Clusiaceae in terms of species richness. In the understory, Rubiaceae,
Araceae, and Piperaceae form the predominant elements (Gentry, 1986). In the lower montane forest, *Cyathea*, Meliaceae (e.g.
*Ruagea*), Fabaceae (e.g. *Dussia*), Melastomataceae (e.g. *Merania, Phainantha*), Rubiaceae (e.g. *Cinchona*), Proteaceae (e.g.
*Roupala*), Lauraceae (e.g. *Nectandra*), and Pteridaceae (e.g. *Pterozonium*) are common elements. Upper montane forests are
dominated *by Myrsine, Ilex, Weinmannia, Clusia, Schefflera, Myrcianthes, Hedyosmum*, and *Oreopanax* (Jørgensen et al.,

132    1999).

Above ca. 3200 m, trees become sparse and eventually the vegetation turns into páramo. The páramo is a unique ecosystem of
the high altitudes of the northern Andes of South America and of southern Central America, located between the continuous
forest line and the permanent snowline at about 3000–5000 m (Luteyn, 1999). The grass páramo is formed by tussock grasses,
mainly *Calamagrostis* and *Festuca*. These are complemented by shrubs of *Diplostephium, Hypericum*, and *Pentacalia*, and
forest patches of *Polylepis*. The shrub páramo consists of cushion plants like *Azorella, Plantago*, and *Werneria*, and shrubs
like *Loricaria* and *Chuquiraga*. The vegetation of the desert páramo is scarce. Some common taxa are *Nototriche, Draba*, and
*Culcitium* (Sklenar and Jorgensen, 1999). The high rates of orographic precipitation that characterize the western part of
equatorial South America cause pollen grains to be washed down by the rain quickly. Therefore, the main transport agent are
the rivers draining into the Pacific. The northern coastal plain of Ecuador is mainly drained by the Esmeraldas and Santiago
Rivers, and the southern coastal plain is drained by several smaller rivers which end in the Guayas River. From the river
mouths, pollen might cross the Peru-Chile Trench in nepheloid layers to reach the Carnegie Ridge.

**1.1.3 Drilling site**

ODP Site 1239 is located at 0°40.32′S, 82°4.86′W, about 120 km offshore Ecuador in a water depth of 1414m, near the eastern
crest of Carnegie Ridge and just next to a downward slope into the Peru-Chile Trench (Mix et al., 2003). Its location is close
to the Equatorial Front (Fig. 2) which separates the warm and low-salinity waters of Panama Basin from the cooler and high-
salinity surface waters of the EEP CT. The region of Site 1239 reveals a thick sediment cover, with dominant sediments in the
region being foraminifer-bearing diatom nannofossil ooze. A tectonic backtrack path on the Nazca plate (Pisias, 1995) reveals
a paleoposition of Site 1239 about 150–200 km further westward (away from the continent) and slightly southward relative to
South America at 4–5 Ma compared to the present day position (Mix et al., 2003). The sediments of Carnegie Ridge are
characterized by high smectite values. Due to its proximity to the Ecuadorian coast, Site 1239 is suitable to record changes in
fluvial runoff, related to variations of precipitation in northwestern South America. Most of the material is discharged by the
Guayas River and Esmeraldas River (Rincon-Martinez et al., 2010).



## 2 Methods

sediment samples of 10 cm³ volume were taken at 67 cm intervals on average from ODP Hole 1239A (cores 33X5-37X1).
Additionally, two core top samples were taken from ODP Hole 1239B as modern analogues. Standard analytical methods were
used to process the samples, including decalcification with HCl (~10%) and removal of silicates with HF (~40%). Two tablets
of exotic *Lycopodium* spores (batch #177,745 containing 18584 ± 829 spores per tablet) were added to the samples during the
decalcification step for calculation of pollen concentrations (grains/cm³). After neutralization with KOH (40%) and washing,
the samples were sieved with ultrasound over an 8μm screen to remove smaller particles. Samples were mounted in glycerin
and a minimum of 100 pollen/spore grains (178 on average) were counted in each sample using a Zeiss Axioskop and 400x
and 1000x (oil immersion) magnification.

For pollen identification, the Neotropical Pollen Database (Bush and Weng, 2007), a reference collection for Neotropical
species held at the Department of Palynology and Climate Dynamics in Göttingen, and related literature (Colinvaux et al.,
1999; Hooghiemstra, 1984; Murillo and Bless, 1974, 1978; Roubik and Moreno, 1991) were used. Pollen types were grouped
according to their main ecological affinity (Flantua et al., 2014; Marchant et al., 2002). The zonation of the diagrams was
based on constrained cluster analysis by sum-of-squares (CONISS), using the square root transformation method (Edwards &
Cavalli–Sforza's chord distance) implemented in TILIA (Grimm, 1991) and visual inspection of the pollen percentage curves.
Percentages are based on the pollen sum which includes all pollen and fern spore types including unidentifiables. Confidence
intervals were calculated after Maher (1972). An initial age model for Site 1239 was established based on biostratigraphic
information (Mix et al., 2003). The age model was refined by matching the benthic stable isotope records from Site 1239 with
those from Site 1241 by visual identification of isotope stages. This procedure resulted in an indirectly orbitally tuned age
model for Site 1239, spanning the interval from 5 to 2.7 Ma (Tiedemann et al., 2007). A hiatus of ca. 5 meters exists between
cores 35X and 36X of Hole 1239A (Tiedemann et al., 2007; Table AT3).

Elemental concentrations (total elemental counts) of Fe and K were measured in high resolution (every 2 cm) using an
Avaatech™ X-Ray Fluorescence (XRF) Core Scanner at the Alfred-Wegener-Institute, Bremerhaven. Both Holes A and B of
ODP Site 1239 were sampled. A nondestructive measuring technique was applied, allowing rapid semi-quantitative
geochemical analysis of sediment cores (Richter et al., 2006). Several studies comparing XRF core scanner data to geochemical
measurements on discrete samples showed that major elements such as Fe, Ca, and K can be precisely measured with the
scanner in a non-destructive way (e.g. Tjallingii et al., 2007).

## 3 Results

Five groups were established with pollen taxa grouped according to their main ecological affinity (Table 1). The groups
páramo, upper montane forest, lower montane forest, and lowland rainforest represent vegetation belts with different altitudinal
ranges (Hooghiemstra, 1984). To track changes of humidity, an additional group named "Indicators of humid conditions" was
established. This group includes those taxa which permanently need humid conditions to grow. Changes of the pollen



percentages of the ecological groups for the Pliocene interval and the core top samples are shown in Fig. 3. Pollen percentages
of single taxa are shown in supplementary Fig. S1. Taxa that occurred in less than 10% of the samples were excluded from the
interpretation.
**3.1 Modern vs. Pliocene pollen assemblages**
Fifty one different palynomorph types were recognized, including 29 pollen and 22 fern spore types. The samples are
characterized by low pollen and spore concentrations of 685 and 465 grains/cm$^3$, respectively. Indicators of humid conditions
show intermediate values. Herbs and grass pollen are very abundant with 20–26%, but tree and shrub pollen decreased to 35–
46% compared to the Pliocene interval. Broad range taxa reach their maximum abundance with 26–27%. Lowland rainforest
and páramo pollen have similar representations as in the Pliocene, whereas the lower and upper montane forest pollen reach
their lowest percentages. When compared to the Pliocene pollen composition, some floristic differences are seen, whereof the
most prominent is the replacement of Podocarpaceae as the most abundant upper montane forest trees by *Alnus*. Another
notable difference is the presence of *Rhizophora* pollen in one of the core top samples, whereas it is completely absent in the
Pliocene interval.

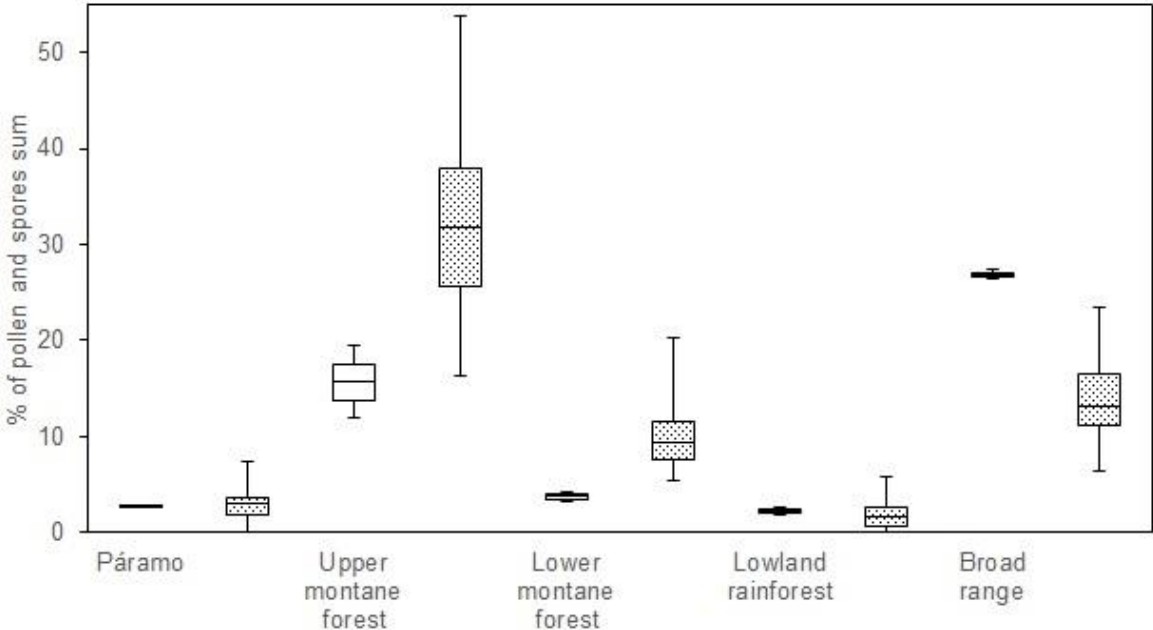


**Figure 3: Comparison of the relative percentages of the different vegetation belts between core top samples (left, plain) and Pliocene**
**samples (right, dotted).**



### 3.2 Description of the Pliocene pollen record

In the Pliocene samples, 141 different palynomorph types were recognized, including 77 pollen and 64 fern spore types. A high percentage of tree and shrub pollen (46–88%) is present throughout the interval, compared to a low percentage of herbs and grass pollen (0–25%; Fig. 4). In most of the vegetation belts, one or two pollen or spore taxa are overrepresented. The lowland rainforest is mainly represented by Polypodiaceae, the lower montane forest is controlled by Cyatheaceae, and the upper montane forest is strongly influenced by Podocarpaceae and *Hedyosmum*. In the páramo, the percentages of the pollen taxa are evenly balanced. Of the total sum, the Andean forest pollen makes by far the largest percentage, with the upper montane forest ranging between 17 and 54% and the lower montane forest between 5 and 19%. The páramo is represented with 0 to 10% and the lowland rainforest with 0 to 6%. The remaining fraction has a wide or unknown ecological range.





**Figure 4: Palynomorph percentages of ODP Hole 1239A for the four vegetation belts and other groups from 4.7 to 4.2 Ma. Grey shading represents the 95% confidence intervals (after Maher, 1972). Vertical black lines delimit the pollen zones. On top elemental ratios of Fe/K from Holes 1239A and 1239B. Ages are from Tiedemann et al. (2007). A hiatus is present in Hole 1239A between 4.45 and 4.55 Ma.**



The pollen record of ODP Hole 1239A was divided into four main pollen zones based on constrained cluster analysis. Pollen
zone I (333.4–323.2 mbsf: 4.70–4.55 Ma, 18 samples) has low pollen and spores concentrations. It is characterized by increases
in pollen values of lowland rainforest, lower montane forest, the percentage of fern spores, and the Fe/K ratio. The pollen
concentrations of broad range taxa, upper montane forest, páramo, and indicators of humid conditions go through frequent
fluctuations. Coastal desert herbs (Amaranthaceae) are well represented. Percentages of Poaceae pollen are low. Between
pollen zone I and II, a hiatus of about 51.7 ka is present. In pollen zone II (319.4–316 mbsf: 4.45–4.42 Ma, 6 samples), the
pollen and spores concentration is similar to pollen zone I. The lowland rainforest pollen, indicators of humid conditions, and
the Fe/K ratio reach their maximum. Fern spores also reach their first maximum. Percentages of lower montane forest and
páramo are high, whereas the percentage of upper montane forest is low at this time due to a strong decline of Podocarpaceae
pollen. The representation of broad range taxa diminish in this interval, the decrease being mainly controlled by *Selaginella*,
Cyperaceae, *Ambrosia/Xanthium*, and Amaranthaceae. Pollen zone III (316–305 mbsf: 4.41–4.26 Ma, 13 samples) shows a
stepwise increase of the pollen and spores concentration with its maximum at 4.3 Ma. The concentration is strongly controlled
by Podocarpaceae pollen which account for up to 44% of the pollen sum in this zone. The pollen of lowland rainforest, lower
montane forest, páramo, indicators of humid conditions, and Fe/K show decreased percentages compared to zone II. Broad
range taxa show some larger fluctuations. The upper montane forest pollen has its maximum extent of this zone (48%) at 4.28
Ma due to the high percentage of Podocarpaceae. If the Podocarpaceae pollen are excluded from the upper montane forest, the
representation of this vegetation belt shows the same pattern of decline as that of the lower montane forest and lowland
rainforest. In pollen zone IV (305–301 mbsf: 4.25–4.2 Ma, 8 samples), the pollen and spores concentration decreases sharply
after 4.24 Ma. The pollen percentage of lower montane forest increases. The percentage of fern spores is at its maximum in
this zone. Percentages of páramo, upper montane forest, broad range taxa, indicators of humid conditions, and the Fe/K ratio
remain similar as in zone III. The percentage of lowland rainforest pollen goes through frequent and large fluctuations.

## 3.3 Description of the páramo

The pollen spectrum from the páramo at ODP Site 1239 includes three different taxa which are mainly confined to the páramo:
the pollen type *Polylepis/Acaena*, and the fern spores *Huperzia* and *Jamesonia/Eriosorus*. Other taxa which are characteristic
of páramos, but cannot be exclusively attributed to this ecosystem because of their broad range occurrence were not included
in the páramo sum (e.g. Asteraceae, Poaceae). The páramo sum constitutes up to 7% of the total pollen and spore sum, with
the highest fractions found at 4.24 and 4.61 Ma, and lowest fractions around 4.23 and 4.59 Ma (Fig. 4). The pollen and spores
types constituting the páramo show similar trends (Fig. 5), which supports the assumption of their common provenance.





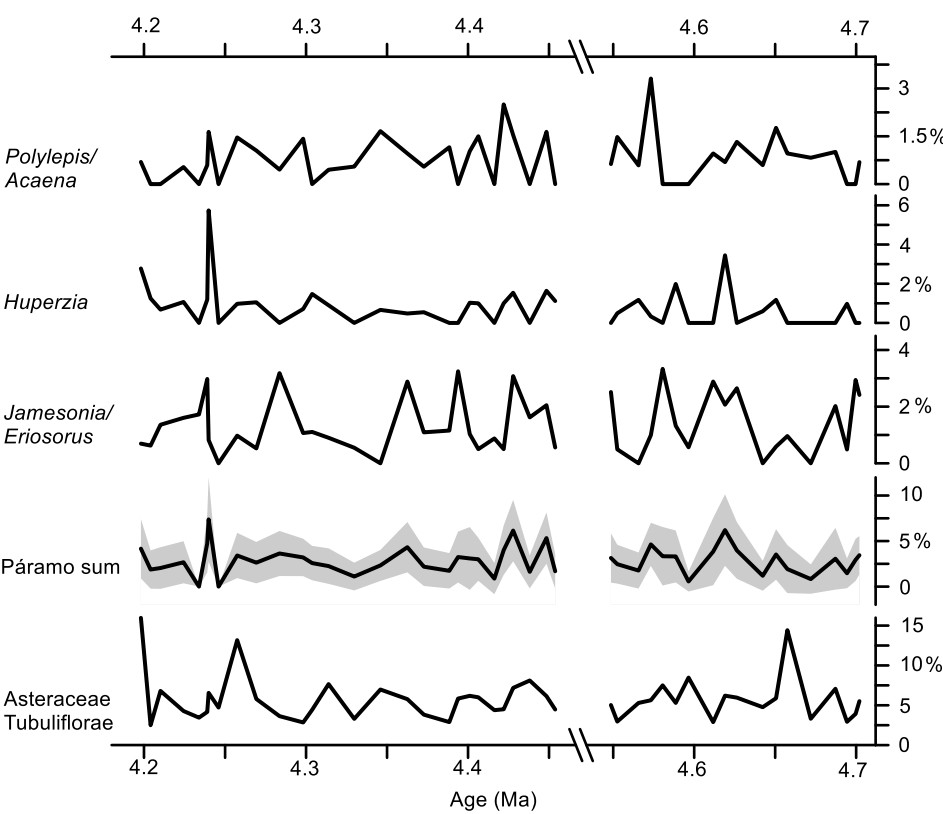

**Figure 5: Development of the páramo. Páramo sum with 95% confidence intervals (grey shading). Asteraceae Tubuliflorae sum excluding *Ambrosia/Xanthium*-type for comparison. Ages are from Tiedemann et al. (2007). Break of axis represents a hiatus between 4.45 and 4.55 Ma.**

## 4 Discussion

### 4.1 The Holocene as modern reference

The core top samples indicate an expansion of broad range taxa and open vegetation, which happened on the expense of the montane forest being strongly diminished compared to the Pliocene situation. This together with the relatively low pollen concentrations would suggest drier conditions. Although there is no detailed age control on these surface/subsurface samples, a Holocene age can be assigned based on the benthic oxygen isotope record (Rincon-Martinez et al., 2010). A Holocene pollen record from nearby core TR 163-38 has high similarity to the core top samples in its youngest part, showing increased open vegetation (Poaceae, Cyperaceae, Asteraceae), low percentages of *Rhizophora*, maximum percentages of fern spores, and low pollen and spores concentrations (González et al., 2006). Despite the expansion of open vegetation, González et al. (2006) interpreted this record to reflect permanently humid conditions, with disturbance processes caused by human occupation and more intense fluvial dynamics. The relatively high percentage of indicators of humid conditions in the core top samples compared to pollen zones III and IV in the Pliocene would be in agreement with this interpretation. The core top samples from

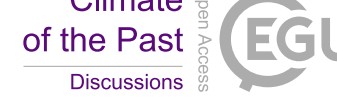



ODP hole 1239B and the most recent part of core TR 163-38 are taken as a basis for the hydrological interpretation of the
Pliocene pollen record.

### 4.2 Climatic implications of vegetation change

The presented marine palynological record provides new information on floristic and vegetation changes occurring along
diverse ecological and climatic gradients through the early Pliocene. The consistently high percentage of tree and shrub pollen,
compared to a low percentage of herbs and grass pollen ($< 25\%$) suggests the predominance of forests and the nearly absence
of open grasslands (below the forest line) during the early Pliocene. Moreover, the very low percentage of dry indicators
(Amaranthaceae) suggests the absence of persisting drought conditions and supports the idea of a rather stable and humid
climate that favored a closed forest cover. This is in good accordance with Pliocene climate models suggesting warmer and
wetter conditions on most continents, which led to expansions of tropical forests and savannas at the expense of deserts
(Salzmann et al., 2011). During the early Pliocene, no profound changes in the vegetation occur. All altitudinal vegetation
belts are already present, with varying ratios. The representation of lowland rainforest goes through the most prominent
development, from being almost absent to about 6% of the pollen sum.
Shifts in vegetation are caused by changes of various parameters such as temperature, precipitation, $CO_2$, radiation, and they
can rarely be explained by a single parameter but are a result of their complex interplay. It is therefore challenging to find the
parameter which has the strongest influence on vegetation. All altitudinal vegetation belts (if Podocarpaceae are excluded)
show a similar pattern of expansion and retreat over time, with an increase in pollen zone I, a maximum in pollen zone II,
retreat in pollen zone III, and another maximum in pollen zone IV. It is known from other Andean pollen records (e.g.
Hooghiemstra and Ran (1994)) that vegetation belts forced by temperature follow a pattern of opposing expansion and retreat.
Such a pattern is seen here in the upper montane forest and the páramo belt (Fig. 5), but the more general pattern which
comprises all vegetation belts seems to reflect changes in hydrology rather than temperature. In this respect, a synchronous
increase/decrease in all vegetation belts is interpreted to reflect more humid/less humid conditions.

### 4.2.1 Development of the coastal vegetation

Pollen zones I and IV show an expansion of coastal desert herbs (Amaranthaceae, Fig. S1), which coincides with low SSTs at
ODP Site 846 in the EEP, suggesting an influence of the Humboldt Current on the coastal vegetation of southern Ecuador.
Remarkably, the lowland rainforest and the coastal desert herbs follow a similar trend. This seems odd at the first glance, but
a possible mechanism to explain this pattern would invoke effects of El Niño. The main transport agent for pollen in this region
are rivers, but in the coastal desert area of southern Ecuador and northern Peru, fluvial discharge rates are low (Milliman and
Farnsworth, 2011). Therefore, pollen might be retained on land until an El Niño event causes severe flooding in the coastal
areas (Rodbell et al., 1999) and episodically fills the rivers which transport the pollen to the ocean. The effects of El Niño seem
to be strongest in pollen zones I and IV where pollen percentages of the lowland rainforest and coastal desert herbs, but also
the upper montane forest, fluctuate most strongly. The lowland rainforest of the coastal plain further north is within the present-



day range of the ITCZ, and expanded from 4.7 Ma onwards possibly due to a southwards displacement of the mean latitude of
the ITCZ (unpublished data from the earliest Pliocene show that the percentage of lowland rainforest before 4.7 Ma was very
low). The development of the lowland rainforest also seems to be related to changes in eustatic sea level. High sea levels
(Miller et al., 2005) coincide with peaks of the lowland rainforest in pollen zones II and IV.
**4.2.2 Development of the montane vegetation**
Podocarpaceae strongly dominate the pollen spectrum in general, but the development of the pollen values is decoupled from
that of all other taxa. This behavior can be explained if a different transport agent is considered. The high pollen production of
Podocarpaceae and their specialized morphology (Regal, 1982) facilitate their eolian transport. In contrast, pollen from most
other taxa is predominantly fluvially transported (González et al., 2006), therefore exhibiting a different pattern where high
pollen concentrations correspond to high fluvial discharge in the source area. The eolian transport of Podocarpaceae explains
the high pollen concentrations in pollen zone III, which occur despite less humid conditions compared to pollen zones II and
IV. The increased eolian transport at 4.63 Ma and between 4.4 and 4.25 Ma is proposed here to be the result of an intensification
of the easterly trade winds. Stronger easterlies also caused the shoaling of the thermocline in the EEP, as shown by models
with a dynamic atmosphere (Zhang et al., 2012). The thermocline in the EEP shoaled between 4.8 and 4.0 Ma (Steph et al.,
2006a, Fig. 6). Related to this process, a critical step of easterly trade wind intensification, indicated by increased eolian
transport of Podocarpaceae pollen, occurred between 4.4 and 4.25 Ma.





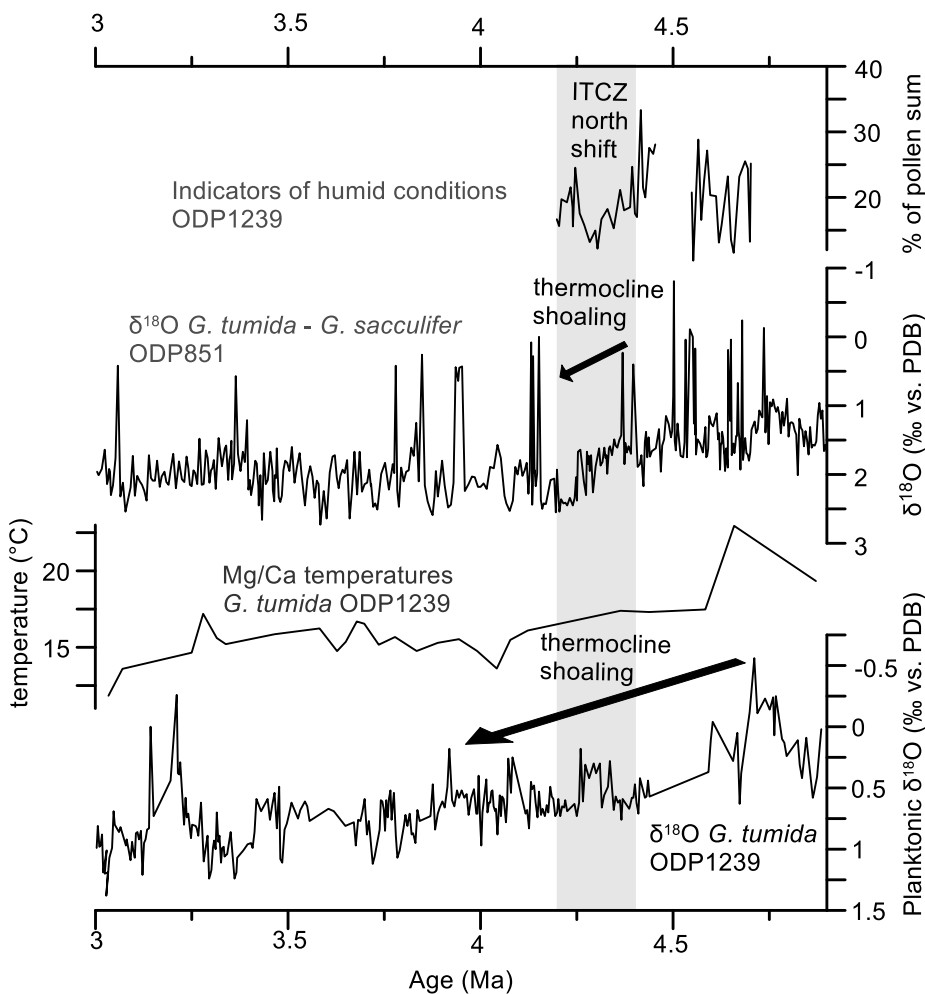

**Figure 6: Percentages of indicators of humid conditions (ODP Site 1239, this study), *G. tumida* – *G. sacculifer* difference in δ¹⁸O from ODP Site 851 in the eastern equatorial Pacific (Cannariato and Ravelo, 1997), and *G. tumida* Mg/Ca temperatures and δ¹⁸O from ODP Site 1239 (Steph, 2005; Steph et al., 2010). Grey shading marks the period of thermocline shoaling at ODP Site 851 and ITCZ north shift.**

Another noteworthy oceanographic change occurred at 4.4 Ma in the EEP. Farrell et al. (1995) described a shift in the locus of maximum opal accumulation rates from ODP Site 850 to ODP Site 846 (Galápagos region), caused by a shift in the availability of nutrients, which is possibly related to increased trade wind strength after 4.4 Ma. Besides being influenced by hydrological changes and wind strength, the upper montane forest and the páramo also respond to temperature changes. Expansions of the upper montane forest combined with retreats of the páramo coincide with higher sea surface temperatures in the EEP (ODP Site 846, Fig. 7). Warmer atmospheric temperatures cause an expansion of the upper montane forest to higher altitudes, resulting in a reduction of the area occupied by páramo and therefore the decline of páramo pollen. On the other hand, higher sea surface temperatures cause higher evaporation and thus higher orographic precipitation in the western Andean Cordillera which might also play a role.



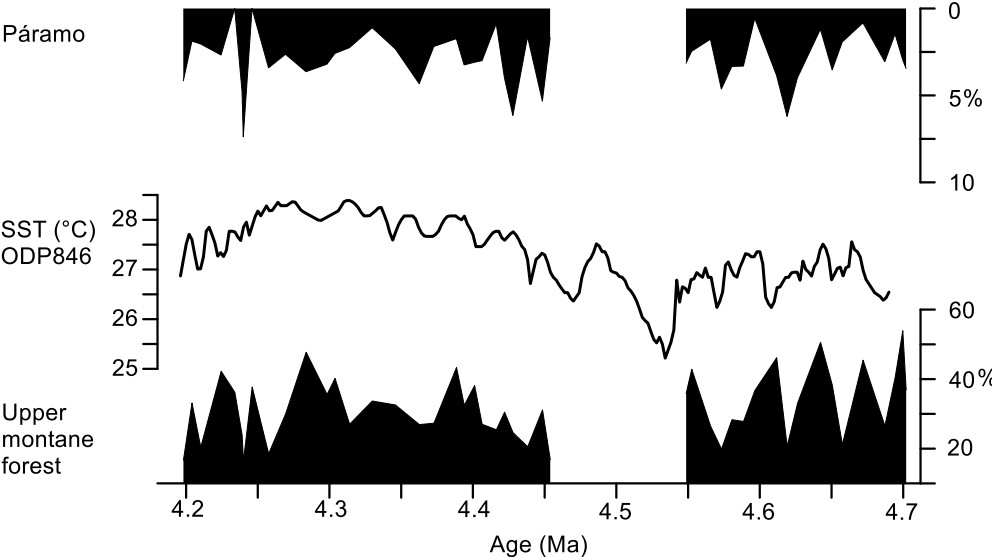

**Figure 7: Sums of upper montane forest and páramo, and UK'37 sea surface temperatures (SST) of ODP site 846 in the eastern equatorial Pacific (Lawrence et al., 2006).**

### 4.3 Fe/K as a tracer for changes in fluvial runoff

The Fe/K ratio has been shown to be a suitable tracer to distinguish terrigenous input of slightly weathered material from drier regions from highly weathered material from humid tropical latitudes. Sediments from deeply chemically weathered terrains have higher iron concentrations compared to the more mobile potassium (Mulitza et al., 2008). Before paleoclimatic interpretations can be made based on elemental ratios, other processes which possibly influence the distribution of Fe/K in marine sediments should be examined, like changes of the topography of Andean river drainage basins, the input of mafic rock material, or diagenetic Fe remobilization (Govin et al., 2012). For northeastern South America it was shown that during the middle Miocene, uplift of the Eastern Andean Cordillera led to changes in the drainage direction of the Orinoco and Magdalena rivers and to the formation of the Amazon River (Hoorn, 1995). If a similar temporal history of uplift and changing drainage patterns is assumed for the western Andean Cordillera, the large-scale patterns of the present topography and river drainage basins should have been in place by the early Pliocene. Therefore, the main direction of fluvial transport of Fe should have been similar to today. Diagenetic alteration was shown not to affect Fe concentrations at Site 1239 (Rincon-Martinez, 2013). The Fe/K ratio therefore seems to be an adequate tracer of fluvial input at this study site. The trend of Fe/K is similar to the pattern of humidity inferred from the pollen spectrum, showing the highest values around 4.46 Ma, thus supporting the hydrological interpretation of the pollen record.

### 4.4 Development of the páramo and implications for Andean uplift

In order to use páramo vegetation as an indicator for Andean elevation, the altitudinal restriction of the páramo taxa to environments above the forest line is a prerequisite. Although no true páramo endemics occurred in the marine samples, or



rather, they could not be identified due to the lack of genus-level morphological distinction (especially *Espeletia* from the
Asteraceae and some Poaceae, e.g. *Festuca*), several taxa are mainly confined to high Andean environments. Dwarf trees of
*Polylepis* typically form patches above the forest line and its natural altitudinal range is thought to occur between a lower limit
which forms the transition to other forest types and up to 5000 m (Kessler, 2002). *Huperzia* occurs in montane forests as
epiphytes and with terrestrial growth form in the páramo (Sklenar et al., 2011). *Jamesonia* and *Eriosorus* are both found in
cool and wet highlands, with most species being found between 2200 and 5000 m (Sanchez and Baracaldo, 2004). Asteraceae
are not restricted to the páramo, but their occurrence in the montane forest and in the lowland rainforest of the Pacific coast is
scarce (Behling et al., 1998). With a contribution of up to 16% of the pollen sum, their source area can be attributed mainly to
the páramo. Additionally, the fluctuations are similar to the other páramo taxa, which is another indication for their common
source area.
The pollen record shows a continuous existence of páramo vegetation without changes in composition. During the warm
Pliocene, the upper montane forest is assumed to have extended to similar or even higher altitudes as today. Despite this
upward expansion of the upper montane forest, the páramo was still present, which implies that the western Cordillera of the
northern Andes had already gone through substantial uplift by that time. Furthermore, the pollen record has a large montane
signature, which would not be the case if the Andes had reached less than half of their modern height by the early Pliocene
(Coltorti and Ollier, 2000). The upper montane forest which constitutes up to 60% of the pollen sum shows that montane
habitats with the corresponding altitudinal belts were already existent. These findings suggest an earlier uplift history for the
western Cordillera of the Northern Andes and according development of the high Andean páramo ecosystem than previously
inferred from palynological studies of the eastern Cordillera in Colombia (Hooghiemstra et al., 2006; Van der Hammen et al.,
1973). This might also be an indication that the uplift history of the western Cordillera of Ecuador is temporally more closely
related to the uplift of the Central Andes where a major phase of uplift occurred between 10 and 6 Ma (Garzione et al., 2008).
In another recent palynological study, the arrival of palynomorphs from the páramo in sediments of the Amazon fan has been
documented since 5.4 Ma (Hoorn et al., 2017). Since the Amazon has its westernmost source in Peru, this signal might be
related to the uplift of the Central Andes. These new records also imply that the modern atmospheric circulation with the Andes
acting as a climate divide has essentially been in place at least since the late Miocene/early Pliocene.
**4.5 Comparing models and proxy data**
Several studies have suggested the existence of a "permanent El Niño" during the Pliocene (e.g. Fedorov et al., 2006; Wara et
al., 2005). El Niño events are characterized by a shift in the Walker circulation, resulting in exceptionally heavy precipitation
particularly over the lowlands of central and southern Ecuador (Bendix and Bendix, 2006) and simultaneous below-average
rainfall over the northwestern slopes of the Andes (Vuille et al., 2000). A permanent El Niño-like climate state during the early
Pliocene would thus have involved permanently humid conditions with high rates of precipitation and fluvial discharge in the
lowlands. Such a climate would have favored the persistence of a broad rain forest coverage and precluded the development
of the desert that exists in coastal southern Ecuador today. The presented pollen record indeed indicates very humid conditions





and the only indicator of dry vegetation is a small percentage of Amaranthaceae pollen. The predicted pattern of expansion of
lowland rainforest at the cost of Andean forest during permanent El Niño is not reflected in the pollen record. Instead, all
altitudinal vegetation belts go through simultaneous shifts of expansion and retreat.
The hypothesis of a permanent El Niño climate state involving a reduced zonal Pacific SST gradient has recently been
questioned as SST reconstructions differ substantially depending on the method. Zhang et al. (2014) claim that a zonal
temperature gradient of ∽3°C has existed since the late Miocene and even intensified during the Pliocene. Our pollen record
instead indicates an influence of periodic El Niño-related variations on the coastal and montane vegetation, especially between
4.7 and 4.55 Ma and between 4.26 and 4.2 Ma, recorded by strong fluctuations in the pollen percentages of coastal and montane
vegetation. The overall parallel expansion and retreat of all vegetation belts would make a more uniform shift in moisture
supply a more likely explanation. Such a shift could be caused by a latitudinal displacement of the ITCZ. A southward
displacement of the ITCZ over both Atlantic and Pacific has been proposed as a response to stronger zonal temperature and
pressure gradients which developed after the restriction of the Central American Seaway and/or a weakening of Southern
Hemisphere temperature gradients (Billups et al., 1999). The timing of the southward shift was narrowed down to 4.4 to 4.3
Ma in this study, based on $\delta^{18}$O records of planktonic foraminifera. The pollen record suggests a slightly different timing, with
a gradual southwards displacement of the ITCZ between 4.7 Ma and 4.42 Ma when the southernmost position was reached. A
less humid phase, indicated by a decrease of humid indicators, lowland rainforest pollen, lower montane forest pollen, and the
Fe/K ratio, followed between 4.42 and 4.26 Ma where the ITCZ presumably had a slightly more northern position. This phase
coincides with the shoaling of the thermocline at ODP Site 851 in the eastern equatorial Pacific (Cannariato and Ravelo, 1997,
Fig. 6). A southward displacement of the ITCZ during the early Pliocene would also be in accordance with eolian deposition
patterns in the EEP which show a latitudinal shift in eolian grain-size and eolian flux between 6 and 4 Ma (Hovan, 1995). The
rather small and slow changes in humidity imply that the ITCZ shift was a gradual process, rather than the response to a single
threshold. Just like the Central American Seaway was restricted and reopened several times before its definitive closure at
around 2.8 Ma (O'Dea et al., 2016), the atmospheric circulation might have adapted gradually in several small steps to these
tectonic changes.
Numerical models suggesting a northward shift of the ITCZ in response to the closure of the Central American Seaway or the
uplift of the northern Andes do not necessarily disagree with an early Pliocene southward shift inferred from proxy data. Both
events occurred gradually over several millions of years and despite recent advances in constraining these events, the timing
of major phases in the uplift histories are still debated. In the case of the Central American Seaway, the timing of surface water
restriction based on diverging salinities in the Caribbean and Pacific ocean, respectively, is well constrained and numerous
global oceanographic changes have been associated with it. Possibly these oceanic reorganizations did not directly trigger
modifications of the atmospheric circulation, but critical periods of uplift influencing atmospheric circulation might have
occurred earlier. On the other hand, the respective model sensitivity experiments generally only consider isolated changes in
single boundary conditions (e.g. closed or open Central American Seaway). Therefore, the effect of those (i.e. a northward
shift of the ITCZ) might counteract the general trend of a southward shift since the late Miocene due to a decrease in the

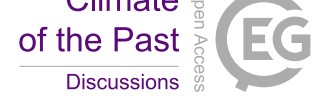



hemispheric temperature gradient (e.g. Pettke et al., 2002). Additionally, global coupled models exhibit uncertainties in the
representation of ocean–atmosphere feedback and cloud–radiation feedbacks, which are especially strong in the study region
(i.e. showing a double ITCZ and an extensive EEP cold tongue (Li and Xie, 2014)). This is problematic also in the light of the
high sensitivity of the ITCZ position to slight shifts in the atmospheric energy balance (Schneider et al., 2014). Another aspect
to consider is that whereas proxy records record the transient response of the climate system over a limited time period, the
mentioned model simulations do not reproduce a stepwise process of environmental changes, e.g. following the closure of the
Panama isthmus (i.e. the shoaling of the thermocline at ~ 4.8 – 4.0 Ma and the start of the EEP cold tongue at ~ 4.3 – 3.6 Ma,
as according to Lawrence et al. (2006) and Steph et al. (2006a)), but the overall equilibrium response.
Concerning the uplift of the northern Andes, there is still a large uncertainty about the time when the Eastern Cordillera reached
its current elevation. Paleobotanists (e.g. Hooghiemstra et al., 2006; Hoorn et al., 2010; Van der Hammen et al., 1973) and
some tectonic geologists (e.g. Mora et al., 2008) argue for a rapid rise of the region since 4–6 Ma, while others conclude that
this is rather unlikely implying an earlier uplift based on biomarker-based paleotemperatures (e.g. Anderson et al., 2015; Mora-
Páez et al., 2016). The estimates for uplift of the Western Cordillera in Ecuador differ even more strongly, and range from
rapid exhumation around 13 and 9 Ma based on thermochronology (Spikings et al., 2005) to a recent uplift during the Pliocene
and Pleistocene (Coltorti and Ollier, 2000). Our pollen record from the páramo shows that the northern Andes must have
already reached close to modern elevations by the early Pliocene. If an early Andean uplift is assumed, the atmospheric
response predicted by the model would have occurred earlier, which would also be in agreement with proxy data indicating a
northern position of the ITCZ during the late Miocene (Hovan, 1995).
Overall, even if the timing and identification of major steps in the shoaling and restriction of the Central American Seaway or
in the uplift of the Northern Andes are resolved, the critical threshold for profound changes in atmospheric circulation and
climate may have occurred at any time during the tectonic processes. Within the analyzed time window, large changes in
atmospheric circulation which have been proposed as a response to the closure of the Central American Seaway (Ravelo et al.,
2004) are absent.

## 5 Conclusions

1)   Between 4.7 and 4.2 Ma, a permanently humid climate with broad rainforest coverage existed in western equatorial

South America. No evidence was found for a permanent El Niño-like climate state, but strong fluctuations in the

vegetation between 4.7 and 4.55 Ma and between 4.26 and 4.2 Ma indicate strong periodic El Niño variability at this

time. Hydrological changes between 4.55 and 4.26 Ma are attributed to gradual shifts of the Intertropical Convergence

Zone which reached its southernmost position around 4.42 Ma and shifted slightly north afterwards.

2)   The most prominent shift in the vegetation occurred in the lowland rainforest.

3)   Between 4.41 and 4.26 Ma, an increased eolian influx of Podocarpaceae pollen indicates an increased strength of the

easterly trade winds, which is presumably related to the shoaling of the EEP thermocline.





4) Results from proxy data and numerical modelling studies regarding the position of the ITCZ during the early Pliocene are not necessarily contradictory. Considering the temporal uncertainties regarding major steps of CAS closure and uplift of the northern Andes, the proposed northward shift of the ITCZ in response to these events might have occurred much earlier (e.g. during the middle to late Miocene).

5) The continuous presence of páramo vegetation implies that by the early Pliocene, the western Cordillera of the northern Andes had already reached an elevation suitable for the development of vegetation above the upper forest line. This new paleobotanical evidence points towards an earlier uplift of the northern Andes than previously suggested by terrestrial paleobotanical records.

Data availability

The underlying research data can be accessed via https://pangaea.de/.

Author contribution

L. Dupont and F. Grimmer conceived the idea, and L. Dupont, F. Grimmer and F. Lamy carried out the analyses. F. Grimmer prepared the manuscript with contributions from all co-authors.

Competing interests

The authors declare that they have no conflict of interest.

**Acknowledgements**

This project was funded by the Deutsche Forschungsgemeinschaft (DFG) through the TROPSAP project (DU221/6) and via the DFG Research Center / Cluster of Excellence "The Ocean in the Earth System — MARUM". The first author thanks GLOMAR – Bremen International Graduate School for Marine Sciences, University of Bremen, Germany, for support. The IODP Gulf Coast Repository (GCR) we acknowledge for their assistance in providing the core samples.

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

| Páramo | Upper montane forest | Lower montane forest | Lowland rainforest | Broad range taxa | Humid indicators |
|---|---|---|---|---|---|
| *Polylepis/ Acaena* | Podocarpaceae | Urticaceae/ Moraceae | Arecaceae | Poaceae | Cyperaceae |
| *Jamesonia/ Eriosorus* | *Hedyosmum* | *Erythrina* | Polypodiaceae | Cyperaceae | *Ranunculus* |
| *Huperzia* | *Clethra* | *Alchornea* | *Pityrogramma-Pteris altissima T* | Tubuliflorae (Asteraceae) | *Hedyosmum* |
| *Ranunculus* *Draba* *Sisyrinchium* | *Myrica* Acanthaceae Melastomataceae | *Styloceras T* Malpighiaceae Cyatheaceae | *Wallichia* | Amaranthaceae Rosaceae *Ambrosia/ Xanthium* | *Ilex* *Pachira* *Myrica* |
| *Cystopteris diaphana T* | *Daphnopsis* | *Vernonia T* | | Ericaceae | Malpighiaceae |
| | *Bocconia* | *Pteris grandifolia T* | | *Artemisia* | Cyatheaceae |
| | *Myrsine* | *Pteris podophylla T* | | *Ilex* | *Selaginella* |
| | *Lophosoria* | *Saccoloma elegans T* | | *Thevetia* | *Pityrogramma-Pteris altissima T* |
| | *Elaphoglossum* | *Thelypteris* | | *Salacia* | *Hymenophyllum T* |
| | *Hypolepis hostilis T* | *Ctenitis subincisa T* | | Bromeliaceae | *Thelypteris* |
| | *Grammitis* | | | Malvaceae | *Ctenitis subincisa T* |
| | *Dodonaea viscosa* | | | Euphorbiaceae | *Alnus* |

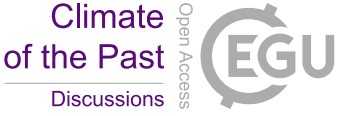



| *Alnus* | | *Liliaceae* | *Cystopteris diaphana T* |
| | | Lycopodiaceae excl. *Huperzia* | |
| | | *Selaginella* | |
| | | *Hymenophyllum T* | |
| | | *Calandrinia* | |
