# Peer review of "Early Pliocene vegetation and hydrology changes in western"

_Climate of the Past, 2017_

## Short Comment (SC1) · 18 Nov 2017

The PAGES Data Stewardship Integrative Activity seeks to advance best practices for sharing the data generated and assembled as part of all PAGES-related activities. The CP Special Issue, "PAGES Young Scientists Meeting 2017" is part of this PAGES activity. The co-editors of the Special Issue are reviewing the data availability within each of the CP-Discussion papers in relation to the CP data policy (https://www.climate-of-the-past.net/about/data_policy.html) and current best practices. The editor team is making recommendations for each paper, with the goal of achieving a high and consistent level of data stewardship across the Special Issue. We recognize that an additional effort will likely be required to meet the high level of data stewardship envisaged, and we appreciate the dedication and contribution of the authors. This includes the use of

[Figure]

Data Citations (see example below). Authors are also strongly encouraged to deposit significant code into a suitable repository and to cite it using a Data Citation.

We ask authors to respond to our comments as part of the regular open interactive discussion. If you have any questions about PAGES Data Stewardship principles, please contact any of us directly. Best wishes for the success of your paper.

YSM Special Issue editor team

Y. Zhang, D.S. Kaufman, H. Plumpton, R. Barnett, M.F. Loutre, M.N. Evans, S.C. Fritz, C. Tabor, E. Razanatsoa, and E. Dearing Crampton Flood

For this paper:

(1) Research input data – proxy and instrumental datasets

This research contribution includes published proxy data in Figures 6-7, including, (i) $\delta$18O record of G. tumida/G. sacculifer at ODP site 851 from Cannariato and Ravelo (1997), (ii) Mg/Ca and $\delta$18O record of G. tumida at ODP Site 1239 from Steph et al. (2005, 2010), (iii) alkenone-based SST record of core ODP 846 (Lawrence et al., 2006), which are already available through existing data repositories. In order to adhere to the Data Policy for Climate of the Past, URLs or full data citations to the primary data must be included in the Data Availability section.

The source of the datasets used to generate Figure 2 are incomplete in the figure caption. Figure 2a: As stated on the NOAA-ESRL website, a bibliographic citation must be included to reference the publication that describes the specific reanalysis product used for the SST field. This is in addition to acknowledging the ESRL for the use of the online data-analysis tool. Figure 2b: Please provide a complete citation for the source of the terrestrial biome map.

(2) Research output data – pollen

This paper presents new and valuable pollen assemblage data from equatorial South

America. In order to adhere to the Data Policy for Climate of the Past, these new data together with (i) the 95% confidence intervals and (ii) the corresponding chronological ages (as showed in Figures 4-5) must be uploaded to an established online data repository (e.g., Neotoma), and a Data Citation or URL link for access to these data must be provided in the Data Availability section of the paper.

What is a "Data Citation"?

Data Citations track the provenance of a dataset giving credit to the data generator; this is in addition to any references to publications where the data are described. Data Citations are used in the text (or tables) alongside and in the same way as publication citations. In the Reference list, they include: Creators, Title, Repository, Identifier, Submission Year. More information about Data Citations is here: <https://www.datacite.org/mission.html> Here is an example of text and corresponding citations (using CP punctuation style):

"The PAGES2k Consortium (2017a) assembled a large global dataset of temperature-sensitive proxy records (PAGES2k Consortium, 2017b). Among the records is the paleo-temperature reconstruction from Laguna Chepical (de Jong et al., 2016), which was described by de Jong et al. (2013)."

References

de Jong, R., von Gunten, I., Maldonado, A., and Grosjean, M.: Late Holocene summer temperatures in the central Andes reconstructed from the sediments of high-elevation Laguna Chepical, Chile (32° S), Climate of the Past, 9, 1921-1932, 2013.

de Jong, R., von Gunten, I., Maldonado, A., and Grosjean, M.: Laguna Chepical summer temperature reconstruction, World Data Center for Paleoclimatology, https://www.ncdc.noaa.gov/paleo/study/20366, 2016.

PAGES 2k Consortium: A global multiproxy database for temperature reconstructions of the Common Era, Scientific Data, 4,170088, 2017a.

PAGES 2k Consortium: A global multiproxy database for temperature reconstructions of the Common Era, version 2.0.0, figshare, https://figshare.com/s/d327a0367bb908a4c4f2, 2017b.

---

## Short Comment (SC2) · 26 Nov 2017

Review Report by Henry Hooghiemstra and Suzette Flantua

Clim. Past Discuss., https://doi.org/10.5194/cp-2017-129

Grimmer, F., Dupont, L., Lamy, F., Jung, G., González, C., Wefer, G., Early Pliocene vegetation and hydrology changes in western equatorial South America.

Background comments on North Andean uplift and terrestrial fossil pollen records: The present paper focusses on the complex history of Andean orogenesis in northwestern South America as seen from marine sediments. An approach from the marine sediment archive is a welcome addition to terrestrial evidence. The lack of well dated sediments in the terrestrial basins fueled much speculation about the uplift history of

the northern Andes, which is strongly related to the closure of the Panamanian Isthmus. Van der Hammen & González (1964) were among the first to speculate how vegetation change could be interpreted in terms of an Andean uplift history.

This discussion was fueled with more data by Van der Hammen et al. (1973). More detailed paleobotanical and palynological evidence was elaborated by Wijninga (1996). A synthesis was published by Hooghiemstra et al. (2006). Indeed, at that time significant uplift of the Colombian Andes was thought to be of Pliocene age a.o. supported by the species-poor páramo (called 'protopáramo') at the base of the Funza-1 pollen record (Hooghiemstra, 1984). The Funza-2 record from the Bogotá basin extended the Funza-1 record back in time, but the age model of the Bogotá sediments continued to be uncertain (Andriessen et al., 1993). A break through in developing an adequate age model was developed by Lucas Lourens (Torres et al. 2013) and the complete Bogotá sediments appeared to reflect the last 2.25 Ma, thus significantly younger as previously thought. In the meantime, mountain building chronology based on apatite fission-track data (Hoorn et al., 2010; see Supplementary Information Fig. S1) showed ages of 30-15 Ma for the most of the Ecuadorian Andes, and ages of 25-10 Ma for most of the Colombian Andes, whereas uplift in specific parts of the Colombian Andes, the Cocuy area in particular, was dated 10-5 Ma.

One may expect that páramo vegetation developed, once a significant proportion of the Andes reached over the elevation of the upper forest line (UFL). Thus, an age of around 10 Ma for the development of the páramo biome is feasible, but so far has been without evidence. In Torres et al. (2013), the species-poor 'proto-paramo' is dated c. 2 Ma. Indeed, molecular phylogenies of high Andean taxa show much and rapid speciation during the Quaternary, mainly driven by the Quaternary ice ages (Van der Hammen, 1974; Diazgranados and Barber, 2017; Pouchon et al., accepted; Flantua and Hooghiemstra, 2017 in Flantua PhD thesis and in book chapter 2018). With the present-day understanding, a much longer history of the development of páramo biome during late Miocene and Pliocene times seems feasible. In absence of strong

climate cycles during pre-Quaternary times (Flantua and Hooghiemstra, in PhD thesis 2017, in book chapter 2018), it may be expected that diversification of the páramo flora increased at low levels during late Miocene and Pliocene times (mainly by migration?; see Baumann, 1988), and speciation shows an explosive increase during the last 2 Ma (see the increasing numbers of studies on molecular phylogenetic trees).

In the absence of long terrestrial records of páramo evolution, pollen analysis of marine sediment cores may provide us with the lacking information. Therefore, the present paper has a fascinating potential to shed new light on this long debated issue. However, I am not surprised that a marine sediment record at this location is only registering weakly the vegetation changes from the adjacent continent. Comparing the regional setting of pollen source areas and operating transport mechanisms in the Eastern Pacific off Ecuador with e.g. the NW African setting (Hooghiemstra et al., 2006), expectations for successful research are relatively poor for the northern Andes. In this respect, the 2006-paper has some predicting power for research in other parts of the world. Notwithstanding potential limitations, the present study is a logical step in exploring the potential of using marine sedimentary archive for insights into continental processes.

Specific comments in text:

line 15: The pollen record presented in this paper reflects the interval 4.7-4.2 Ma only in 46 pollen samples. There is no justification why not a longer interval has been analysed.

18: The sediment core is located at the equator and one may wonder which area is considered as 'study area'. Fig. 2b offers the possibility to better show the potential pollen transport routes from the various source areas to site ODP 1239. We guess that pollen source areas stretch more to the south than to the north.

30: Montes 2015 should be Montes et al. 2015. If you really want to capture the discussion around the closure, then you should already add here: O'Dea A, Aguilera O, Aubry M-P, Berggren WA, Budd AF, Cione AL, Coates AG, Collins LS, Coppard SE,

Cozzuol MA, de Queiroz A, Duque-Caro H, Eytan RI, Farris DW, Finnegan S, Gasparini GM, Grossman EL, Johnson KG, Keigwin LD, Knowlton N, Leigh EG, Leonard-Pingel JS, Lessios HA, Marko PB, Norris RD, Rachello-Dolmen PG, Restrepo-Moreno SA, Soibelzon E, Soibelzon L, Stallard RF, Todd JA, Vermeij GJ, Woodburne MO, Jackson JBC. 2016a. Formation of the Isthmus of Panama. Science Advances. e1600883.' A. O'Dea, H. A. Lessios, A. G. Coates, R. I. Eytan, S. A. Restrepo-Moreno, A. L. Cione, L. S. Collins, A. de Queiroz, D. W. Farris, R. D. Norris, R. F. Stallard, M. O. Woodburne, O. Aguilera, M.-P. Aubry, W. A. Berggren, A. F. Budd, M. A. Cozzuol, S. E. Coppard, H. Duque-Caro, S. Finnegan, G. M. Gasparini, E. L. Grossman, K. G. Johnson, L. D. Keigwin, N. Knowlton, E. G. Leigh, J. S. Leonard-Pingel, P. B. Marko, N. D. Pyenson, P. G. Rachello-Dolmen, E. Soibelzon, L. Soibelzon, J. A. Todd, G. J. Vermeij, J. B. C. Jackson. Building bridges. Response to Erkens and Hoorn: "The Panama Isthmus, 'old', 'young' or both?". Science Advances 2016b: Vol. 2, no. 8, e1600883 DOI: 10.1126/sciadv.1600883

33. See also figure 1 from O'Dea et al. 2016b

57: 'Northern Andes' needs to be better specified as the uplift history in Ecuador differs from Colombia.

63: Additionally to Spikings et al. 2005: see Supplementary Information Fig. S1 in Hoorn et al., 2010.

74: first palynological record = first marine palynological record. Pollen records of Pliocene age are Rio Frio-17 and Subachoque-39 (Wijninga, 1996 figures 8.2 and 8.4; Hooghiemstra et al., 2006 figures 4 and 7) that show the floral composition of montane forest and páramo. Table 4 of the 2006-paper may be helpful in considering how pollen taxa are grouped to reflect the altitudinal vegetation zones.

81: make a difference between the 'Western Cordillera' in Ecuador and the 'Western Cordillera' in Colombia (see also lines 62-63).

85: Figure 2b shows 'terrestrial biomes' in 7 legend units. However, this palynological study is using 4 legend units (the left hand four in Table 1). The link between pollen source areas and pollen diagram can be improved if for consistency Figure 2b would show the same categories as recognized in Table 1. Fig. 2b can be improved by indicating with arrows the routes and mechanisms (ocean currents, rivers, wind?) of pollen and spore transport towards ODP 1239. The information of the inset figure can be placed (with less detail) in the main figure.

95: The acronym ENSO has been introduced already in line 49.

96: Marchant et al., (2001) is a good example of small-scale climate patterns.

118: Here you specify "the mountains" as the area within Ecuador, while before you talk broadly about the "Northern Andes". For consistency, the Ecuadorean Andes/Cordilleras should be used throughout the paper.

127: Palmae = Arecaceae

141: Santiago river not shown in map?

140-141 & 162: The combination of pollen and fern spores in the pollen sum is not ideal. In Lowland Forest (< 1000 m) and Lower Montane Forest fern-loaded trees fallen across small rivers shed enormous amounts of spores directly into the water currents (personal observations). Therefore, the proportion of fern spores in the pollen spectra are of a different character than the proportion of pollen, and both components are difficult to compare (see also figures 72-75 in Hooghiemstra et al., 1986).

143: It would be very useful to have figure 2 show these geographical features such as the Carnegie Ridge and Peru-Chile Trench, and show how pollen would reach the location of the record.153-154: What is the potential pollen supply by the Humbold Current ? and where could pollen source areas be located? Furthermore, from this description it's not clear why you are expecting to find signals from the eastern lowlands which constitute the western margin of the Amazon Basin (line 118).

156: Avoid starting the sentence with a number. See 191.

183 / Table 1: (a) Table 1 shows an effort to group pollen and spore taxa into meaningful ecological groups. A marine pollen record has a wide pollen source area and many taxa listed do have a wide ecological range. The latter makes it difficult to develop clear-cut ecological groups. Páramo: Several 'páramo' taxa also occur in the forest; Upper Montane Forest: Melastomataceae from the UMF also occur abundantly in the páramo; Lowland Forest (LF): is represented by a remarkably low number of taxa; Broad-range taxa: Poaceae indeed have a broad ecological range, but most of it comes from páramo and also dry vegetation in coastal areas and interandean valleys; Asteraceae Tubuliflorae also have a wide range but the bulk is from the páramo; Artemisia seems from Peruvian origin?; Thevetia is indicator of dry conditions rather than a broad range plant; Humid indicators: the following 4 taxa are advised to be omitted from this list: Ranunculaceae (important plant in the puna), Ilex (indeed, also in wet forest but frequently elsewhere), Myrica (= Morella), and Malpighiaceae (also in savannas).

(b) A significant proportion of Lowland Forest (including wet rainforest and dry deciduous forest) and Lower Montane Forest is palynologically 'silent' as many trees are pollinated by insects, beetles, etc. Given the northward flow of the Humboldt Current, and atmospheric circulation in the direction of the coast, I am not surprised that LF is hardly represented by taxa in Table 1. Palms are more abundant in Lower Montane forest (LMF); by moving the Arecaceae from the category LF to LMF, LF is hardly reflected at all. This has consequences for the interpretation and conclusions.

(c) Most of the taxa in the category 'páramo' occur in the ecotone zone of the UMF. Thus the record for 'paramo' as based on the taxa listed in Table 1 may also reflect the zone with dwarf forest and shrub. (d) In Flantua et al. (2014, figure 7) we explored how altitudinal vegetation zonation is reflected on the basis of GBIF data. We were quite disappointed with the poor altitudinal zonation, possibly explained by a large amount of 'noise' in the GBIF data by using data from a wide geographical region. We concluded

that up to date an altitudinal zonation based on expert knowledge from field botanists gives better results: see Groot et al., 2013 (RPP) figure 3).

185: Here, make also reference to Van der Hammen (1974).

188: Which taxa occurred in less than 10% ? A Table showing all identified pollen and spore taxa and their assessment to ecological categories would be helpful.

192: to what refers 'respectively'?

194: 26-27% ? Does this relate to a figure?

197: Where is the replacement of Podocarpus by Alnus shown in a figure?

207: 'The Lowland Rainforest is mainly represented by Polypodiaceae'. However, Polypodiaceae actually occur everywhere and are not a representative of lowland rainforest in particular. Figure 4: curves are easier to read with a horizontal line at 0%. The elevation indicated for páramo and UFL overlap by 100 m. Is this on purpose? Can your modern samples be shown here as well for comparison?

217-237: make reference to the figure where all these changes can be seen.

229: 'of the pollen sum' is redundant.

232-233: Podocarpus excluded from the UMF; if I understand well, not from the pollen sum.

240: 2 out of the 3 species mentioned also occur in the forest (Polylepis, Huperzia). The species mentioned are not a strong indication of the presence of páramo. Lycopodium with foveolate spores is most characteristic of páramo vegetation (Van 't Veer et al., 1992); absence of Lycopodium fov. in the pollen spectra is in support of the view that the present taxa identified as 'reflecting páramo' also are reflecting lowermost páramo and ecotone forest.

245: Fig. 5: where is the curve showing páramo vs. montane forest? 'páramo sum' is

a confusing term; better to use 'paramo taxa (%)'

251-253: difficult to understand why the evidence mentioned is suggesting drier conditions. Please explain more clearly.

255: better to show the location of TR 163-38 on the map in Fig. 2.

267: difficult to understand the claim '(below the forest line)' . How can open grassland below and above the UFL be identified and separated from each other? This seems an over-interpretation of the data.

270: 'expansion of savannas' ? whereas (line 266) there is a near absence of open grasslands.

272: 'All altitudinal vegetation belts are already present'. Most possibly correct but not necessarily in its present form. For example, after Quercus had changed the composition of montane forest (LMF and UMF) several LMF taxa were able to reach higher elevations (Hooghiemstra, 1984; Torres et al., 2013). Unfortunately, modern climatological constraints of the lower and upper boundaries of the LMF are insufficiently understood (Hooghiemstra et al., 2012); as a consequence it is difficult to infer climatological change from altitudinal migrations of LMF.

272: 'goes through the most prominent' is unclear, please rephrase.

278: 'It is known from other Andean pollen records ....' The comparison made here should be better explained.

277: 'show a similar pattern of expansion': what do you mean exactly?, and where can the reader see this expansion?

279-282: unclear text, needs rephrasing. For instance, unclear use of 'opposing', what pattern is exactly to be seen in the fig. 5, what is the "more general pattern"?

287 'the main transport agent for pollen' ; I guess also for spores? Replace here and in the following sentences "El Niño" by ENSO.

292: it would be useful to have a figure that can support this statement on 'lowland rainforest of the coastal plain further north' as it's unclear what "further north" is. Or indicate with lat/long values.

294: 'lowland rainforest' is poorly reflected by the taxa listed in Table 1. and as a consequence it is difficult to make a comparison.

298: Difficult to understand what means 'the development of pollen values is decoupled from'. Needs a better explanation and visualization.

300 'eolian' transport is contra to line 140-141.

317: 'Besides being influenced by hydrological changes and wind strength' is unclear and needs further explanation.

321: Replace western Andean Cordillera with western Cordillera of Ecuador. Be consistent throughout the text with Western or western.

324: Sums of upper montane forest = Representation (%) of upper montane forest

334: better to use the more recent reference '(Hoorn et al., 2010)'

333+335: Add 'Ecuador' to Eastern and Western Andean Cordillera

342: In order to use páramo vegetation = In order to use the abundance of páramo vegetation

343: Replace 'no true páramo endemics' by 'Although no taxa restricted to páramo only were identified. . ..'

347: 'Polylepis is reaching 5000 m in the northern Andes': I guess this refers to Peru and Bolivia and maximum elevations relate to individual trees. In Colombia and Ecuador Polylepis dwarf forest occurs up to 4200-4300 m.

349-351: Perhaps not as present in montane forest and lowland rainforest, but relatively close to your marine record, you have the presence of several major forest nuclei of

seasonally dry tropical forest biome (see Särkinen et al. 2011) and there are a number of different species of Asteraceae in Peruvian seasonally dry tropical forest (see book Neotropical Savannas and Seasonally Dry Forests: Plant Diversity, Biogeography, and Conservation by T. Pennington & J. Ratter 2006). Could this biome be the source of Asteraceae in your record?

354: 'without changes in composition' is rather meaningless as so few páramo taxa have been identified.

355: which evidence is fueling this assumption?

356-357: the weak evidence of páramo does not allow to infer conclusions about the elevation of the Andes.

362-368: uplift histories of the various areas are confusing here: 362: indeed uplift is older as can be seen in Hoorn et al., 2010.

364: uplift of the Central Andes is 60-25 Ma (instead of 10-6 Ma; see Hoorn et al., 2010, Suppl. Info. )

365: Amazon fan = Amazon Fan

365: Which is the first palynological paper to state here "in another recent palynological study.."?

366: The Hoorn et al. 2017 paper suggests but does not provide conclusive evidence that the grass pollen are from páramo as the source area for the Amazon river include also high Andean open vegetation of the puna. This sentence here should be rephrased to not 'oversell' Hoorn et al. 2017 in support of páramo presence.

377: Amaranthaceae and Thevetia rather are reflecting dry conditions.

379: what is the meaning of 'all altitudinal vegetation belts go through simultaneous shifts of expansion and retreat' ?

[Figure]

382: Add space before the 3.

385: Better explain 'parallel expansion and retreat of all vegetation belts'. For the last 20 ka we have learned that little goes parallel (see Hooghiemstra and Van der Hammen, 2004).

419: Eastern Cordillera reached = Eastern Cordillera of Colombia reached

421: 'argue for a rapid rise of the region since 4-6 Ma' ; This is outdated and should be 30-5 Ma (see Hoorn et al., 2010 Suppl. Info.)

425: 'Our pollen record from the páramo shows .....' This conclusion seems unwarranted as the evidence for páramo vegetation is weak and also could reflect ecotone forest and/or other biomes.

435: On which evidence is this sentence based?

440: Conclusion 2 is difficult to understand: when? a shift to what?

441: Higher representation of Podocarpaceae is interpreted as evidence of more intense trade winds. However, this is not necessarily the case as pollen record Funza09 (Torres et al., 2013, figure 10) shows that Podocarpus is more abundant during several intervals of Pleistocene time, potentially also leading to high representation in the marine sediments.

447-448: The presence of páramo is weakly supported by evidence; the inferred altitude of the Ecuadorian (?) Andes is speculative as a consequence.

449-450: Better to refer to more recent literature in which the uplift of the Northern Andes has been set back in time already.

564: Reference Montes et al. 2015 is incomplete.

Fig. S1: To which degree modern core top samples are comparable to the pre-Quaternary samples? Are mechanisms of pollen transport comparable? Some remarks about this issue are missing.

Fig. S2: % sum páramo = páramo (place the word 'percentage' in the figure caption) : also for other taxa

Fig. S3: Mention in the caption 'Pollen percentage diagram' and omit all % % indications on top of the pollen diagram. And: Myrica = Morella

IN CONCLUSION:

* The biomes 'páramo' and 'lowland rainforest' are hardly reflected by characteristic pollen and spore taxa. Several taxa now classified as 'broad range taxa' cold be shifted to 'páramo' but with the same restriction that these taxa also could reflect uppermost montane forest (ecotone forest).

* In marine pollen records changing proportions of pollen taxa / ecological groups may reflect vegetation change and / or changes in pollen transport. In the present manuscript the latter is hardly/not considered (This remark also relates to the suggestions for improvement of Fig. 2).

* Integration of terrestrial and marine proxies is a powerful tool to maximize conclusions. The comparison with model output has broadened the scope of this paper but – apart from speculation - has not generated an incremental step forwards.

* Pollen zones in Fig. S3 are not expressive and the interpretation in terms of environmental change is not convincing. The presented pollen evidence does not allow a full support of the suggested conclusions of this paper. Analysing a much longer interval has the potential to strengthen conclusions, but the regional setting will remain poor to obtain convincing evidence.

References mentioned in the report:

Andriessen, P.A.M., Helmens, K.F., Hooghiemstra, H., Riezebos, P.A., Van der Hammen, T., 1993. Absolute chronology of the Pliocene-Quaternary sediment sequence of

the Bogotá area, Colombia. Quaternary Science Reviews 12, 483-501.

Baumann, F., 1988. Geographische Verbreitung und Okologie südamerikanischer Hochgebirgspflanzen. . Beitrag zur Rekonstruktion der quartären Vegetations-geschichte der Anden. Physische Geographie 28, 206 pp. Geographische Institut der Universität, Zürich.

Diazgranados, M., Barber, J.C., 2017. Geography shapes the phylogeny of frailejones (Espeletiinae Cuatrec., Asteraceae): a remarkable example of recent rapid radiation in sky islands. PeerJ 5, e2968. https://doi.org/10.7717/peerj.2968

Flantua, S.G.A., Hooghiemstra, H., 2017. Unravelling the mountain fingerprint: topography, paleoclimate and connectivity as drivers of contemporary biodiversity pattern in the Northern Andes. In: Flantua, S.G.A. (ed.), Climate change and topography as drivers of Latin American biome dynamics. PhD thesis, University of Amsterdam, pp. 265-307.

Flantua, S.G.A., Hooghiemtra, H., 2018. Historical connectivity and mountain biodiversity. In: Hoorn, C., Parrigo, A., Antonelli, A. (eds.), Mountains, climate and biodiversity. Wiley, Chichester, UK.

Groot, M.H.M., Hooghiemstra, H., Berrio, J.-C., Giraldo, C., 2013. North Andean environmental and climatic change at orbital to submillennial time-scales: vegetation, water-levels, and sedimentary regimes from Lake Fúquene during 130-27 ka. Review of Palaeobotany and Palynology 197, 186-204.

Hooghiemstra, H., 1984. Vegetational and climatic history of the high plain of Bogota, Colombia: a continuous record of the last 3.5 million years. Dissertationes Botanicae 79, 368 pp.

Hooghiemstra, H., Van der Hammen, T., 2004. Quaternary ice-age dynamics in the Colombian Andes: developing an understanding of our legacy. Philosophical Transactions of the Royal Society London B 359, 173-181.

Hooghiemstra, H., Agwu, C.O.C., Beug, H.-J., 1986. Pollen and spore distribution in recent marine sediments: a record of NW-African seasonal wind patterns and vegetation belts. Meteor' Forschungs Ergebnisse, Reihe C 40, 87-135.

Hooghiemstra, H., Lézine, A.-M., Leroy, S.A.G., Dupont, L., Marret, F., 2006. Late Quaternary palynology in marine sediments: a synthesis of the understanding of pollen distribution patterns in the NW African setting. Quaternary International 148, 29-44.

Hoorn, C., Wesselingh, F.P., ter Steege, H., Bermudez, M.A., Mora, A., Sevink, J., Sanmartín, I., Sanchez-Meseguer, A., Anderson, C.L., Figueiredo, J.P., Jaramillo, J., Riff, D., Negri, F.R., Hooghiemstra, H., Lundberg, J., Stadler, T., Särkinen, T., Antonelli, A., 2010. Amazonia through time: Andean uplift, climate change, landscape evolution, and biodiversity. Science 330, 927-931.

Marchant, R., Behling, H., Berrio, J.C., Cleef, A., Duivenvoorden, J., Hooghiemstra, H., Kuhry, P., Melief, B., Van Geel, B, Van der Hammen, T., Van Reenen, G., Wille, M., 2001. Mid- to late-Holocene pollen-based biome reconstructions for Colombia. Quaternary Science Reviews 20, 1289-1308.

Torres, V., Hooghiemstra, H., Lourens, L.J., Tzedakis, P.C. 2013. Astronomical tuning of long pollen records reveals the dynamic history of montane biomes and lake levels in the tropical high Andes during the Quaternary. Quaternary Science Reviews 63, 59-72.

Van der Hammen, T. & González, E. (1964). A pollen diagram from th Quaternary of the Sabana de Bogota (Colombia) and its significance for the geology of the northern Andes. Geologie en Mijnbouw 43, 113-117.

Van der Hammen, T., Werner, J.H. & Van Dommelen, H., 1973. Palynological record of the upheaval of the Northern Andes; a study of the Pliocene and Lower Quaternary of the Colombian Eastern Cordillera and the early evolution of its high-Andean biota. Palaeogeography Palaeoclimatology Palaeoecology 16, 1-122.

[Figure]

Van 't Veer, R., Ran, E.T.H., Mommersteeg, H.J.P.M., Hooghiemstra, H., 1995. Multivariate analysis of the middle and late Pleistocene Funza pollen record of Colombia. Mededelingen Rijks Geologische Dienst 52, 195-212.

Wijninga, V.M., 1996. Paleobotany and palynology of Neogene sediments from the high plain of Bogotá (Colombia): evolution of the Andean flora from a paleoecological perspective. PhD thesis, University of Amsterdam, The Netherlands, 370 pp.

Henry Hooghiemstra, Suzette Flantua

University of Amsterdam

26 November 2017

---

## Referee Comment (RC1) · C. Hoorn (Referee) · 21 Dec 2017

Paper: Early Pliocene vegetation and hydrology changes in western equatorial South America by Grimmer et al.

Reviewer Carina Hoorn

Summary The purpose of the paper is to establish the direction of shift of the ITCZ following the closure of the Central American Seaway (CAS) and uplift of the northern Andes. The paper comprises a palynological study of sediments from the interval between 4.7 and 4.2 Ma of the appropriately situated ODP core 1239A. The specific aims are to reconstruct vegetation, climate and topography in this region throughout this time interval. The conclusion is that an (already) high Andean landscape existed

at the time, and that both vegetation and landscape during this interval match with a scenario corresponding to a southward shift of the ITCZ. Fluctuations of the ENSO are also considered. The results are in accordance with other paleoceanographic data in the region.

Main comments:

There is a shortage of continuous records from the Pliocene in the eastern Pacific that reflect hydrological and climatic change in the region. This paper aims to fill this gap. However, the dataset makes it hard to see the big changes that one would expect from the text. If possible the dataset should be extended with additional data to which are referred in the text.

• The interaction of Andean uplift, closure of the CAS, shifting ITCZ and ENSO altogether make it quite a daunting task to interpret the palynological diagram an assign changes to specific causes. The case is clearly made and looked at from all angles. Question: Is there a chance that some of the subtle changes in the diagram can be related to the Pliocene uplift pulses in the Andes and related atmospheric changes? Such pulses are postulated in tectonic reconstructions (e.g. Anderson et al., 2015, Geosphere) and are mentioned by authors in the paragraph starting at line 464.

• The new dataset further confirm that a high topography (Anderson et al., 2015) and paramo (Bermudez et al., 2015 in Basin Research; Hoorn et al., 2017 in Global & Planetary Change) was in place at least since the early Pliocene. It might be worthwhile highlighting the regional character of this condition?

Note that modern type precipitation patterns are likely to have been in place already from middle Miocene onwards (see Kaandorp et al., 2006; Hoorn et al., 2010; Barnes et al., 2012) and this would have required a significant orographic barrier. A high Andes might go as far back as the mid-Miocene, however, first evidence for a paramo is now set as latest Miocene to early Pliocene. Lines 406-407 could be reconsidered in this context.

⢠The elemental concentrations analysis needs to be better introduced and is currently rather hidden and makes a surprise first appearance in the methods section. In methods also explain why this is a useful additional technique. Part of the text in section 4.3 (line 360 onwards) could be moved to the introduction to explain approach.

⢠The discussion of the Holocene samples in relation to the Pliocene seems a bit ambivalent and does not form a very good guideline to better understand the new results.

⢠Lines from 313: A rather crucial line comes up here and reads as follows: "unpublished data from the earliest Pliocene show that the percentage of lowland rainforest before 4.7 Ma was very low". The evidence that is presented seems rather subtle and perhaps not iconic for an important vegetation & climate change. The authors allude to data of the earliest Pliocene, which they say strengthen their case. However, they are not visible. If these data belong to the authors it might be timely to include them here (or a selection of them) and make a more compelling case.

⢠A map with the scenarios for the changing ITCZ would be welcome. Instead this could also be added to figure 1.

Minor comments:

⢠In line 465 Hoorn et al. 2010 are listed as backing up a rapid rise of the region since 4– 6 Ma, However we suggest in the mid-Miocene the Andes must have already been high with further uplift at a later stage.

⢠The writing style at places can be somewhat convolute and could do with rephrasing. A suggestion for the opening sentence would be: "The progressive closure of the Central American Seaway (CAS) and the uplift of the northern Andes profoundly reorganized early Pliocene ocean and atmospheric circulation in the Eastern Equatorial Pacific (EEP)."

---

## Short Comment (SC3) · 15 Mar 2018

This study generated new vegetation and climate record between 4.7 and 4.2 Ma by pollen analysis of 46 samples from ODP Site 1239A, which is located in the East Equatorial Pacific, a place suitable for investigating the precipitation-related fluvial runoff changes in northwestern South America, thus good for monitoring the past movement of the ITCZ. A major aim of this study is to clarify a mismatch about the ITCZ shift in the early Pliocene between the proxy records and the model simulation, that most proxy data supports a southward shift whereas numerical modelling suggests a northward shift in response to Central American Seaway closure and Andean uplift. Generally, this study fills the blank of well dated hydrological record of the early Pliocene in this region by pollen and spores studies from marine sediment.

[Figure]

Generally, I agree with the comments posted by the other three referees and won't repeat it. Here are some minor suggestions, which I think should help the readers to better understand this research if considered.

Age model. How did the authors establish the age model for the study interval of Site 1239? From Tiedemann et al. (2007)? Why not add the benthic d18O record to the figures and sign labels of Marine Isotope Stages? You cannot just cite a reference to get all the necessary things done.

Continuity of the record. Since other palynological studies of the region have been conducted for the mid-Pliocene to the Holocene, why not combine those records with the new record of the early Pliocene? Are they from the same marine core? The new record depends on 46 samples to cover the time interval of 4.7-4.2 Ma, with an average time resolution of 11 Kyr. In such a relatively short period and with a relatively low time resolution, the authors still recognize four major steps of the vegetation changes, and claim that all the vegetation belts as explained in Figures 3 and showed in Figure 4 display synchronous increase/decrease for each stage. If carefully examining figure 4, the features of the variability of the vegetation belts just constrainedly match those described in the text. The referee RC1 also pointed it out. Increasing the time resolution such as doubling, and filling the hiatus between cores 35X and 36X of Hole 1239A (there should be also vegetation change in this interval), something very different probably could happen. Also as indicated by Referee RC1, the unpublished data which is so important to support the author's conclusion of a low percentage of lowland rainforest before 4.7 Ma should be put together with the presented record of this manuscript. I believe that all readers with interests for the ITCZ shift in the early Pliocene would like to see a continuous record since the early Pliocene rather than a segmented record in such a narrow period.

About permanent Elño, closure of Central American Seaway and Andean uplift. My suggestion is weakening the discussion on these comprehensive topics but focusing on its significance in indicting the hydrological changes. The new pollen records are not

strong evidences to support the so ambitious conclusions in the present manuscript.

---

## Author Comment (AC2) · 26 Jun 2018

**Figure 2.** Modern climate (boreal summer) and vegetation and core site positions of ODP Sites 677, 846, 851, 1000, 1239, 1241, Trident core TR163-38, and M772-056 mentioned in the text. **A**. Long-term monthly July precipitation in mm/day (CPC) and wind field (NCEP). July is the middle of the rainy season in northern South America, when the ITCZ is at its northern boreal summer position. Salinity estimates for the Caribbean indicate a position of the ITZC further north during the Pliocene. Direction of wind is not favorable for wind transport of pollen and spores to ODP Site 1239. **B**. Long-term monthly July sea surface temperatures (NODC), surface and subsurface currents of the eastern equatorial Pacific (Mix et al. 2003). NECC, North Equatorial Countercurrent; SEC, South Equatorial Current; PCC, Peru-Chile Current (continuation of the Humboldt Current); CC, Coastal Current; EUC, Equatorial Undercurrent; GUC, Gunther Undercurrent. **C**. Contours, bathymetry (ETOPO1), main rivers in Ecuador, and vegetation. Transport of pollen and spores in the ocean over the Peru-Chile Trench, which is very narrow east of the Carnegie Ridge, probably takes place in nepheloid layers. Páramo vegetation is found between 3200 and 4800 m, upper montane Andean forest (UMF) grows between 1000 and 2300 m, sub-Andean lower montane forest (LMF) between 1000 and 2300 m, and lowland forest (LR) below 1000m. The distribution of desert and xeric shrubs in northern Peru, drier broad-leaved forest, flooded grasslands, and mangroves in Ecuador and Colombia is denoted in different colors (see legend, WWF). Source areas of pollen and spores in sediments of ODP Site 1239 are sought in western Ecuador, northwestern Peru, and southwestern Colombia (see text).

References:
Mix, Tiedemann, Blum et al. 2003: Init Report ODP, leg 202.
World Wildlife Fund (WWF). https://www.worldwildlife.org/ecoregions/ (retrieved November 2017)
National centers for Environmental Information (NOAA):
– CPC Merged Analysis of Precipitation. http://www.cdc.noaa.gov/cdc/data.cmap.html (retieved February 2008)
– ETOPO1. https://maps.ngdc.noaa.gov/viewers/wcs-client/ (retrieved June 2018)
– NCEP reanalysis data (meridional and zonal wind). http://www.esrl.noaa.gov/psd/data/gridded/data.ncep.reanalysis.derived.html (retrieved January 2010)
– NODC (Levitus) World Ocean Atlas.http://www.esrl.noaa.gov/psd/data/gridded/data.nodc.woa94.html (retrieved March 2010)

[Figure]

**Figure 3.** Comparison of the palynomorph percentages (based on total pollen and spores) of Podocarpaceae and the different vegetation belts between 2 Holocene samples (black) and Pliocene samples between 4.7-4.2 Ma (box-whisker plots).

[Figure]

**Figure 4A (extra figure).** Pliocene and Pleistocene palynomorph percentages (based on the total of pollen and spores) of ODP Hole 1239A for three vegetation belts, humidity indicators, grass pollen and pollen and spore concentration per ml. 95% confidence intervals as grey bars after Maher (1972).

**Additional Results**. Percentages of humidity indicators hint to slightly drier conditions at the beginning of the Pliocene. A trend towards higher palynomorph concentrations is found for the period from 6 to 2 Ma. Grass pollen percentages remain low indicating mainly closed forest at altitudes below the Páramo. Representation of lowland rainforest was low around 4.7 Ma, increased by 4.5 Ma, declined again to low levels around 3.5 Ma, and rose to remain at higher levels during the Pleistocene. Continuous presence of pollen and spores from the Páramo indicates that the northern Andes had reached high altitudes in Ecuador before the Pliocene.

[Figure]

**Figure 4.** Palynomorph percentages of ODP Hole 1239A for the four vegetation belts and other groups from 4.7 to 4.2 Ma. Grey shading represents the 95% confidence intervals (after Maher, 1972). Vertical black lines delimit the pollen zones. At the top stable oxygen isotopes of the benthic foraminifer *C. wuellerstorfi* (Tiedemann et al., 2007) of ODP Hole 1239A, marine isotope stages (MIS), and elemental ratios of Fe/K from Holes 1239A and 1239B. Ages are from Tiedemann et al. (2007). A coring gap is present in Hole 1239A between 4.45 and 4.55 Ma.

[Figure]

**Figure 5.** Palynomorph percentages of Páramo indicators and Asteraceae Tubuliflorae (excluding Ambrosia/Xanthium T) of the past 6 Ma indicating the presence of Páramo vegetation at least since the late Miocene. 95% confidence intervals (grey bars) after Maler (1972). Ages after Tiedemann et al. (2007).

Age (ka)

[Figure]

% pollen + spores

Age (Ma)

**Table 1.** List of identified pollen and spore taxa in marine ODP Holes 1239A (Pliocene samples) and 1239B 683 (Holocene samples, taxa in grey occurred only these samples) and grouping according to their main ecological affinity (Flantua et al., 2014; Marchant et al., 2002).

| Páramo | Upper montane forest | Lower montane forest | Lowland rainforest | Broad range taxa | Humid indicators |
|---|---|---|---|---|---|
| *Polylepis/ Acaena* | Podocarpaceae | Urticaceae/ Moraceae | *Wettinia* | Poaceae | Cyperaceae |
| *Jamesonia/ Eriosorus* | *Hedyosmum* | *Erythrina* | *Socratea* | Cyperaceae | *Ranunculus* |
| *Huperzia* | *Clethra* | *Alchornea* | Polypodiaceae | Tubuliflorae (Asteraceae) | *Hedyosmum* |
| *Ranunculus* | *Morella* | *Styloceras T* | *Pityrogramma-Pteris altissima T* | Amaranthaceae | *Ilex* |
| *Draba* | Acanthaceae | Malpighiaceae | | Rosaceae | *Pachira* |
| *Sisyrinchium* | Melastomataceae | Cyatheaceae | | *Ambrosia/ Xanthium* | *Myrica* |
| *Cystopteris diaphana T* | *Daphnopsis* | *Vernonia T* | | Ericaceae | Malpighiaceae |
| | *Bocconia* | *Pteris grandifolia T* | | *Artemisia* | Cyatheaceae |
| | *Myrsine* | *Pteris podophylla T* | | *Ilex* | *Selaginella* |
| | *Lophosoria* | *Saccoloma elegans T* | | *Thevetia* | *Pityrogramma-Pteris altissima T* |
| | *Elaphoglossum* | *Thelypteris* | | *Salacia* | *Hymenophyllum T* |
| | *Hypolepis hostilis T* | *Ctenitis subincisa T* | | Bromeliaceae | *Thelypteris* |
| | *Grammitis* | | | Malvaceae | *Ctenitis subincisa T* |
| | *Dodonaea viscosa* | | | Euphorbiaceae | *Alnus* |
| | *Alnus* | | | *Liliaceae* | *Cystopteris diaphana T* |
| | | | | Lycopodiaceae excl. *Huperzia* | |
| | | | | *Selaginella* | |
| | | | | *Hymenophyllum T* | |
| | | | | *Calandrinia* | |

[Figure]

Supplementary figure. Pollen percentage diagram against age (Tiedemann et al., 2007), with total counts, percentages of single taxa and groups, pollen zones, CONISS clusters. On top two samples from the Holocene. Minor ticks denote 1%, major ticks 2%, unless stated differently. This panel shows pollen and spore taxa from mangrove, lowland rainforest and lower montane forest. Panels on the next page show the pollen percentages for taxa from the upper montane forest, Páramo, and broad range taxa.

[Figure]

Supplementary figure (continued)

---

## Author Response (AR1)

The PAGES Data Stewardship Integrative Activity seeks to advance best practices for sharing the data generated and assembled as part of all PAGES-related activities. The CP Special Issue, "PAGES Young Scientists Meeting 2017" is part of this PAGES activity. The co-editors of the Special Issue are reviewing the data availability within each of the CP-Discussion papers in relation to the CP data policy (https://www.climate-of-the-past.net/about/data_policy.html) and current best practices. The editor team is making recommendations for each paper, with the goal of achieving a high and consistent level of data stewardship across the Special Issue. We recognize that an additional effort will likely be required to meet the high level of data stewardship envisaged, and we appreciate the dedication and contribution of the authors. This includes the use of

Data Citations (see example below). Authors are also strongly encouraged to deposit significant code into a suitable repository and to cite it using a Data Citation.

We ask authors to respond to our comments as part of the regular open interactive discussion. If you have any questions about PAGES Data Stewardship principles, please contact any of us directly. Best wishes for the success of your paper.

YSM Special Issue editor team

Y. Zhang, D.S. Kaufman, H. Plumpton, R. Barnett, M.F. Loutre, M.N. Evans, S.C. Fritz, C. Tabor, E. Razanatsoa, and E. Dearing Crampton Flood

For this paper:

(1) Research input data – proxy and instrumental datasets

This research contribution includes published proxy data in Figures 6-7, including, (i) $\delta$18O record of G. tumida/G. sacculifer at ODP site 851 from Cannariato and Ravelo (1997), (ii) Mg/Ca and $\delta$18O record of G. tumida at ODP Site 1239 from Steph et al. (2005, 2010), (iii) alkenone-based SST record of core ODP 846 (Lawrence et al., 2006), which are already available through existing data repositories. In order to adhere to the Data Policy for Climate of the Past, URLs or full data citations to the primary data must be included in the Data Availability section.

The source of the datasets used to generate Figure 2 are incomplete in the figure caption. Figure 2a: As stated on the NOAA-ESRL website, a bibliographic citation must be included to reference the publication that describes the specific reanalysis product used for the SST field. This is in addition to acknowledging the ESRL for the use of the online data-analysis tool. Figure 2b: Please provide a complete citation for the source of the terrestrial biome map.

(2) Research output data – pollen

This paper presents new and valuable pollen assemblage data from equatorial South

America. In order to adhere to the Data Policy for Climate of the Past, these new data together with (i) the 95% confidence intervals and (ii) the corresponding chronological ages (as showed in Figures 4-5) must be uploaded to an established online data repository (e.g., Neotoma), and a Data Citation or URL link for access to these data must be provided in the Data Availability section of the paper.

What is a "Data Citation"?

Data Citations track the provenance of a dataset giving credit to the data generator; this is in addition to any references to publications where the data are described. Data Citations are used in the text (or tables) alongside and in the same way as publication citations. In the Reference list, they include: Creators, Title, Repository, Identifier, Submission Year. More information about Data Citations is here: <https://www.datacite.org/mission.html> Here is an example of text and corresponding citations (using CP punctuation style):

"The PAGES2k Consortium (2017a) assembled a large global dataset of temperature-sensitive proxy records (PAGES2k Consortium, 2017b). Among the records is the paleo-temperature reconstruction from Laguna Chepical (de Jong et al., 2016), which was described by de Jong et al. (2013)."

References de Jong, R., von Gunten, I., Maldonado, A., and Grosjean, M.: Late Holocene summer temperatures in the central Andes reconstructed from the sediments of high-elevation Laguna Chepical, Chile (32° S), Climate of the Past, 9, 1921-1932, 2013.

de Jong, R., von Gunten, I., Maldonado, A., and Grosjean, M.: Laguna Chepical summer temperature reconstruction, World Data Center for Paleoclimatology, https://www.ncdc.noaa.gov/paleo/study/20366, 2016.

PAGES 2k Consortium: A global multiproxy database for temperature reconstructions of the Common Era, Scientific Data, 4,170088, 2017a.

PAGES 2k Consortium: A global multiproxy database for temperature reconstructions of the Common Era, version 2.0.0, figshare, https://figshare.com/s/d327a0367bb908a4c4f2, 2017b.

Clim. Past Discuss.,
https://doi.org/10.5194/cp-2017-129-AC3, 2018

[Figure]

Data are uploaded in Pangaea as datasets PANGAEA.884280 and PANGAEA.884153, which are combined in PANGAEA.884285. We update and complete these datasets (upload in progress) with the data from the new supplementary figure.

Clim. Past Discuss.,
https://doi.org/10.5194/cp-2017-129-SC2, 2017

[Figure]

Grimmer, F., Dupont, L., Lamy, F., Jung, G., González, C., Wefer, G., Early Pliocene vegetation and hydrology changes in western equatorial South America.

Background comments on North Andean uplift and terrestrial fossil pollen records: The present paper focusses on the complex history of Andean orogenesis in northwestern South America as seen from marine sediments. An approach from the marine sediment archive is a welcome addition to terrestrial evidence. The lack of well dated sediments in the terrestrial basins fueled much speculation about the uplift history of the northern Andes, which is strongly related to the closure of the Panamanian Isthmus. Van der Hammen & González (1964) were among the first to speculate how vegetation change could be interpreted in terms of an Andean uplift history.

This discussion was fueled with more data by Van der Hammen et al. (1973). More detailed paleobotanical and palynological evidence was elaborated by Wijninga (1996). A synthesis was published by Hooghiemstra et al. (2006). Indeed, at that time significant uplift of the Colombian Andes was thought to be of Pliocene age a.o. supported by the species-poor páramo (called 'protopáramo') at the base of the Funza-1 pollen record (Hooghiemstra, 1984). The Funza-2 record from the Bogotá basin extended the Funza-1 record back in time, but the age model of the Bogotá sediments continued to be uncertain (Andriessen et al., 1993). A break through in developing an adequate age model was developed by Lucas Lourens (Torres et al. 2013) and the complete Bogotá sediments appeared to reflect the last 2.25 Ma, thus significantly younger as previously thought. In the meantime, mountain building chronology based on apatite fission-track data (Hoorn et al., 2010; see Supplementary Information Fig. S1) showed ages of 30-15 Ma for the most of the Ecuadorian Andes, and ages of 25-10 Ma for most of the Colombian Andes, whereas uplift in specific parts of the Colombian Andes, the Cocuy area in particular, was dated 10-5 Ma.

One may expect that páramo vegetation developed, once a significant proportion of the Andes reached over the elevation of the upper forest line (UFL). Thus, an age of around 10 Ma for the development of the páramo biome is feasible, but so far has been without evidence. In Torres et al. (2013), the species-poor 'proto-paramo' is dated c. 2 Ma. Indeed, molecular phylogenies of high Andean taxa show much and rapid speciation during the Quaternary, mainly driven by the Quaternary ice ages (Van der Hammen, 1974; Diazgranados and Barber, 2017; Pouchon et al., accepted; Flantua and Hooghiemstra, 2017 in Flantua PhD thesis and in book chapter 2018). With the present-day understanding, a much longer history of the development of páramo biome during late Miocene and Pliocene times seems feasible. In absence of strong climate cycles during pre-Quaternary times (Flantua and Hooghiemstra, in PhD thesis 2017, in book chapter 2018), it may be expected that diversification of the páramo flora increased at low levels during late Miocene and Pliocene times (mainly by migration?; see Baumann, 1988), and speciation shows an explosive increase during the last 2 Ma (see the increasing numbers of studies on molecular phylogenetic trees).

In the absence of long terrestrial records of páramo evolution, pollen analysis of marine sediment cores may provide us with the lacking information. Therefore, the present paper has a fascinating potential to shed new light on this long debated issue. However, I am not surprised that a marine sediment record at this location is only registering weakly the vegetation changes from the adjacent continent. Comparing the regional setting of pollen source areas and operating transport mechanisms in the Eastern Pacific off Ecuador with e.g. the NW African setting (Hooghiemstra et al., 2006), expectations for successful research are relatively poor for the northern Andes. In this respect, the 2006-paper has some predicting power for research in other parts of the world. Notwithstanding potential limitations, the present study is a logical step in exploring the potential of using marine sedimentary archive for insights into continental processes.

Specific comments in text:

line 15: The pollen record presented in this paper reflects the interval 4.7-4.2 Ma only in 46 pollen samples. There is no justification why not a longer interval has been analysed.

18: The sediment core is located at the equator and one may wonder which area is considered as 'study area'. Fig. 2b offers the possibility to better show the potential pollen transport routes from the various source areas to site ODP 1239. We guess that pollen source areas stretch more to the south than to the north.

30: Montes 2015 should be Montes et al. 2015. If you really want to capture the discussion around the closure, then you should already add here: O'Dea A, Aguilera O, Aubry M-P, Berggren WA, Budd AF, Cione AL, Coates AG, Collins LS, Coppard SE,

Cozzuol MA, de Queiroz A, Duque-Caro H, Eytan RI, Farris DW, Finnegan S, Gasparini GM, Grossman EL, Johnson KG, Keigwin LD, Knowlton N, Leigh EG, Leonard-Pingel JS, Lessios HA, Marko PB, Norris RD, Rachello-Dolmen PG, Restrepo-Moreno SA, Soibelzon E, Soibelzon L, Stallard RF, Todd JA, Vermeij GJ, Woodburne MO, Jackson JBC. 2016a. Formation of the Isthmus of Panama. Science Advances. e1600883.' A. O'Dea, H. A. Lessios, A. G. Coates, R. I. Eytan, S. A. Restrepo-Moreno, A. L. Cione, L. S. Collins, A. de Queiroz, D. W. Farris, R. D. Norris, R. F. Stallard, M. O. Woodburne, O. Aguilera, M.-P. Aubry, W. A. Berggren, A. F. Budd, M. A. Cozzuol, S. E. Coppard, H. Duque-Caro, S. Finnegan, G. M. Gasparini, E. L. Grossman, K. G. Johnson, L. D. Keigwin, N. Knowlton, E. G. Leigh, J. S. Leonard-Pingel, P. B. Marko, N. D. Pyenson, P. G. Rachello-Dolmen, E. Soibelzon, L. Soibelzon, J. A. Todd, G. J. Vermeij, J. B. C. Jackson. Building bridges. Response to Erkens and Hoorn: "The Panama Isthmus, 'old', 'young' or both?". Science Advances 2016b: Vol. 2, no. 8, e1600883 DOI: 10.1126/sciadv.1600883

33. See also figure 1 from O'Dea et al. 2016b

57: 'Northern Andes' needs to be better specified as the uplift history in Ecuador differs from Colombia.

63: Additionally to Spikings et al. 2005: see Supplementary Information Fig. S1 in Hoorn et al., 2010.

74: first palynological record = first marine palynological record. Pollen records of Pliocene age are Rio Frio-17 and Subachoque-39 (Wijninga, 1996 figures 8.2 and 8.4; Hooghiemstra et al., 2006 figures 4 and 7) that show the floral composition of montane forest and páramo. Table 4 of the 2006-paper may be helpful in considering how pollen taxa are grouped to reflect the altitudinal vegetation zones.

81: make a difference between the 'Western Cordillera' in Ecuador and the 'Western Cordillera' in Colombia (see also lines 62-63).

85: Figure 2b shows 'terrestrial biomes' in 7 legend units. However, this palynological study is using 4 legend units (the left hand four in Table 1). The link between pollen source areas and pollen diagram can be improved if for consistency Figure 2b would show the same categories as recognized in Table 1. Fig. 2b can be improved by indicating with arrows the routes and mechanisms (ocean currents, rivers, wind?) of pollen and spore transport towards ODP 1239. The information of the inset figure can be placed (with less detail) in the main figure.

95: The acronym ENSO has been introduced already in line 49.

96: Marchant et al., (2001) is a good example of small-scale climate patterns.

118: Here you specify "the mountains" as the area within Ecuador, while before you talk broadly about the "Northern Andes". For consistency, the Ecuadorean Andes/Cordilleras should be used throughout the paper.

127: Palmae = Arecaceae

141: Santiago river not shown in map?

140-141 & 162: The combination of pollen and fern spores in the pollen sum is not ideal. In Lowland Forest (< 1000 m) and Lower Montane Forest fern-loaded trees fallen across small rivers shed enormous amounts of spores directly into the water currents (personal observations). Therefore, the proportion of fern spores in the pollen spectra are of a different character than the proportion of pollen, and both components are difficult to compare (see also figures 72-75 in Hooghiemstra et al., 1986).

143: It would be very useful to have figure 2 show these geographical features such as the Carnegie Ridge and Peru-Chile Trench, and show how pollen would reach the location of the record.153-154: What is the potential pollen supply by the Humbold Current ? and where could pollen source areas be located? Furthermore, from this description it's not clear why you are expecting to find signals from the eastern lowlands which constitute the western margin of the Amazon Basin (line 118).

156: Avoid starting the sentence with a number. See 191.

/ Table 1: (a) Table 1 shows an effort to group pollen and spore taxa into meaningful ecological groups. A marine pollen record has a wide pollen source area and many taxa listed do have a wide ecological range. The latter makes it difficult to develop clear-cut ecological groups. Páramo: Several 'páramo' taxa also occur in the forest; Upper Montane Forest: Melastomataceae from the UMF also occur abundantly in the páramo; Lowland Forest (LF): is represented by a remarkably low number of taxa; Broad-range taxa: Poaceae indeed have a broad ecological range, but most of it comes from páramo and also dry vegetation in coastal areas and interandean valleys; Asteraceae Tubuliflorae also have a wide range but the bulk is from the páramo; Artemisia seems from Peruvian origin?; Thevetia is indicator of dry conditions rather than a broad range plant; Humid indicators: the following 4 taxa are advised to be omitted from this list: Ranunculaceae (important plant in the puna), Ilex (indeed, also in wet forest but frequently elsewhere), Myrica (= Morella), and Malpighiaceae (also in savannas).

(b) A significant proportion of Lowland Forest (including wet rainforest and dry deciduous forest) and Lower Montane Forest is palynologically 'silent' as many trees are pollinated by insects, beetles, etc. Given the northward flow of the Humboldt Current, and atmospheric circulation in the direction of the coast, I am not surprised that LF is hardly represented by taxa in Table 1. Palms are more abundant in Lower Montane forest (LMF); by moving the Arecaceae from the category LF to LMF, LF is hardly reflected at all. This has consequences for the interpretation and conclusions.

(c) Most of the taxa in the category 'páramo' occur in the ecotone zone of the UMF. Thus the record for 'paramo' as based on the taxa listed in Table 1 may also reflect the zone with dwarf forest and shrub. (d) In Flantua et al. (2014, figure 7) we explored how altitudinal vegetation zonation is reflected on the basis of GBIF data. We were quite disappointed with the poor altitudinal zonation, possibly explained by a large amount of 'noise' in the GBIF data by using data from a wide geographical region. We concluded that up to date an altitudinal zonation based on expert knowledge from field botanists gives better results: see Groot et al., 2013 (RPP) figure 3).

185: Here, make also reference to Van der Hammen (1974).

188: Which taxa occurred in less than 10% ? A Table showing all identified pollen and spore taxa and their assessment to ecological categories would be helpful.

192: to what refers 'respectively'?

194: 26-27% ? Does this relate to a figure?

197: Where is the replacement of Podocarpus by Alnus shown in a figure?

207: 'The Lowland Rainforest is mainly represented by Polypodiaceae'. However, Polypodiaceae actually occur everywhere and are not a representative of lowland rainforest in particular. Figure 4: curves are easier to read with a horizontal line at 0%. The elevation indicated for páramo and UFL overlap by 100 m. Is this on purpose? Can your modern samples be shown here as well for comparison?

217-237: make reference to the figure where all these changes can be seen.

229: 'of the pollen sum' is redundant.

232-233: Podocarpus excluded from the UMF; if I understand well, not from the pollen sum.

240: 2 out of the 3 species mentioned also occur in the forest (Polylepis, Huperzia). The species mentioned are not a strong indication of the presence of páramo. Lycopodium with foveolate spores is most characteristic of páramo vegetation (Van 't Veer et al., 1992); absence of Lycopodium fov. in the pollen spectra is in support of the view that the present taxa identified as 'reflecting páramo' also are reflecting lowermost páramo and ecotone forest.

245: Fig. 5: where is the curve showing páramo vs. montane forest? 'páramo sum' is a confusing term; better to use 'paramo taxa (%)'

251-253: difficult to understand why the evidence mentioned is suggesting drier conditions. Please explain more clearly.

255: better to show the location of TR 163-38 on the map in Fig. 2.

267: difficult to understand the claim '(below the forest line)' . How can open grassland below and above the UFL be identified and separated from each other? This seems an over-interpretation of the data.

270: 'expansion of savannas' ? whereas (line 266) there is a near absence of open grasslands.

272: 'All altitudinal vegetation belts are already present'. Most possibly correct but not necessarily in its present form. For example, after Quercus had changed the composition of montane forest (LMF and UMF) several LMF taxa were able to reach higher elevations (Hooghiemstra, 1984; Torres et al., 2013). Unfortunately, modern climatological constraints of the lower and upper boundaries of the LMF are insufficiently understood (Hooghiemstra et al., 2012); as a consequence it is difficult to infer climatological change from altitudinal migrations of LMF.

272: 'goes through the most prominent' is unclear, please rephrase.

278: 'It is known from other Andean pollen records ....' The comparison made here should be better explained.

277: 'show a similar pattern of expansion': what do you mean exactly?, and where can the reader see this expansion?

279-282: unclear text, needs rephrasing. For instance, unclear use of 'opposing', what pattern is exactly to be seen in the fig. 5, what is the "more general pattern"?

'the main transport agent for pollen' ; I guess also for spores? Replace here and in the following sentences "El Niño" by ENSO.

292: it would be useful to have a figure that can support this statement on 'lowland rainforest of the coastal plain further north' as it's unclear what "further north" is. Or indicate with lat/long values.

294: 'lowland rainforest' is poorly reflected by the taxa listed in Table 1. and as a consequence it is difficult to make a comparison.

298: Difficult to understand what means 'the development of pollen values is decoupled from'. Needs a better explanation and visualization.

'eolian' transport is contra to line 140-141.

317: 'Besides being influenced by hydrological changes and wind strength' is unclear and needs further explanation.

321: Replace western Andean Cordillera with western Cordillera of Ecuador. Be consistent throughout the text with Western or western.

324: Sums of upper montane forest = Representation (%) of upper montane forest

334: better to use the more recent reference '(Hoorn et al., 2010)'

333+335: Add 'Ecuador' to Eastern and Western Andean Cordillera

342: In order to use páramo vegetation = In order to use the abundance of páramo vegetation

343: Replace 'no true páramo endemics' by 'Although no taxa restricted to páramo only were identified. . ..'

347: 'Polylepis is reaching 5000 m in the northern Andes': I guess this refers to Peru and Bolivia and maximum elevations relate to individual trees. In Colombia and Ecuador Polylepis dwarf forest occurs up to 4200-4300 m.

349-351: Perhaps not as present in montane forest and lowland rainforest, but relatively close to your marine record, you have the presence of several major forest nuclei of seasonally dry tropical forest biome (see Särkinen et al. 2011) and there are a number of different species of Asteraceae in Peruvian seasonally dry tropical forest (see book Neotropical Savannas and Seasonally Dry Forests: Plant Diversity, Biogeography, and Conservation by T. Pennington & J. Ratter 2006). Could this biome be the source of Asteraceae in your record?

354: 'without changes in composition' is rather meaningless as so few páramo taxa have been identified.

355: which evidence is fueling this assumption?

356-357: the weak evidence of páramo does not allow to infer conclusions about the elevation of the Andes.

362-368: uplift histories of the various areas are confusing here: 362: indeed uplift is older as can be seen in Hoorn et al., 2010.

364: uplift of the Central Andes is 60-25 Ma (instead of 10-6 Ma; see Hoorn et al., 2010, Suppl. Info. )

365: Amazon fan = Amazon Fan

365: Which is the first palynological paper to state here "in another recent palynological study.."?

366: The Hoorn et al. 2017 paper suggests but does not provide conclusive evidence that the grass pollen are from páramo as the source area for the Amazon river include also high Andean open vegetation of the puna. This sentence here should be rephrased to not 'oversell' Hoorn et al. 2017 in support of páramo presence.

377: Amaranthaceae and Thevetia rather are reflecting dry conditions.

379: what is the meaning of 'all altitudinal vegetation belts go through simultaneous shifts of expansion and retreat' ?

382: Add space before the 3.

385: Better explain 'parallel expansion and retreat of all vegetation belts'. For the last 20 ka we have learned that little goes parallel (see Hooghiemstra and Van der Hammen, 2004).

419: Eastern Cordillera reached = Eastern Cordillera of Colombia reached

421: 'argue for a rapid rise of the region since 4-6 Ma' ; This is outdated and should be 30-5 Ma (see Hoorn et al., 2010 Suppl. Info.)

425: 'Our pollen record from the páramo shows .....' This conclusion seems unwarranted as the evidence for páramo vegetation is weak and also could reflect ecotone forest and/or other biomes.

435: On which evidence is this sentence based?

440: Conclusion 2 is difficult to understand: when? a shift to what?

441: Higher representation of Podocarpaceae is interpreted as evidence of more intense trade winds. However, this is not necessarily the case as pollen record Funza09 (Torres et al., 2013, figure 10) shows that Podocarpus is more abundant during several intervals of Pleistocene time, potentially also leading to high representation in the marine sediments.

447-448: The presence of páramo is weakly supported by evidence; the inferred altitude of the Ecuadorian (?) Andes is speculative as a consequence.

449-450: Better to refer to more recent literature in which the uplift of the Northern Andes has been set back in time already.

564: Reference Montes et al. 2015 is incomplete.

Fig. S1: To which degree modern core top samples are comparable to the pre-Quaternary samples? Are mechanisms of pollen transport comparable? Some re- marks about this issue are missing.

Fig. S2: % sum páramo = páramo (place the word 'percentage' in the figure caption) : also for other taxa

Fig. S3: Mention in the caption 'Pollen percentage diagram' and omit all % % indications on top of the pollen diagram. And: Myrica = Morella

IN CONCLUSION:

* The biomes 'páramo' and 'lowland rainforest' are hardly reflected by characteristic pollen and spore taxa. Several taxa now classified as 'broad range taxa' cold be shifted to 'páramo' but with the same restriction that these taxa also could reflect uppermost montane forest (ecotone forest).

* In marine pollen records changing proportions of pollen taxa / ecological groups may reflect vegetation change and / or changes in pollen transport. In the present manuscript the latter is hardly/not considered (This remark also relates to the suggestions for improvement of Fig. 2).

* Integration of terrestrial and marine proxies is a powerful tool to maximize conclusions. The comparison with model output has broadened the scope of this paper but – apart from speculation - has not generated an incremental step forwards.

* Pollen zones in Fig. S3 are not expressive and the interpretation in terms of environmental change is not convincing. The presented pollen evidence does not allow a full support of the suggested conclusions of this paper. Analysing a much longer interval has the potential to strengthen conclusions, but the regional setting will remain poor to obtain convincing evidence.

References mentioned in the report:

Andriessen, P.A.M., Helmens, K.F., Hooghiemstra, H., Riezebos, P.A., Van der Hammen, T., 1993. Absolute chronology of the Pliocene-Quaternary sediment sequence of the Bogotá area, Colombia. Quaternary Science Reviews 12, 483-501.

Baumann, F., 1988. Geographische Verbreitung und Okologie südamerikanischer Hochgebirgspflanzen. . Beitrag zur Rekonstruktion der quartären Vegetations-geschichte der Anden. Physische Geographie 28, 206 pp. Geographische Institut der Universität, Zürich.

Diazgranados, M., Barber, J.C., 2017. Geography shapes the phylogeny of frailejones (Espeletiinae Cuatrec., Asteraceae): a remarkable example of recent rapid radiation in sky islands. PeerJ 5, e2968. https://doi.org/10.7717/peerj.2968

Flantua, S.G.A., Hooghiemstra, H., 2017. Unravelling the mountain fingerprint: topography, paleoclimate and connectivity as drivers of contemporary biodiversity pattern in the Northern Andes. In: Flantua, S.G.A. (ed.), Climate change and topography as drivers of Latin American biome dynamics. PhD thesis, University of Amsterdam, pp. 265-307.

Flantua, S.G.A., Hooghiemtra, H., 2018. Historical connectivity and mountain biodiversity. In: Hoorn, C., Parrigo, A., Antonelli, A. (eds.), Mountains, climate and biodiversity. Wiley, Chichester, UK.

Groot, M.H.M., Hooghiemstra, H., Berrio, J.-C., Giraldo, C., 2013. North Andean environmental and climatic change at orbital to submillennial time-scales: vegetation, water-levels, and sedimentary regimes from Lake Fúquene during 130-27 ka. Review of Palaeobotany and Palynology 197, 186-204.

Hooghiemstra, H., 1984. Vegetational and climatic history of the high plain of Bogota, Colombia: a continuous record of the last 3.5 million years. Dissertationes Botanicae 79, 368 pp.

Hooghiemstra, H., Van der Hammen, T., 2004. Quaternary ice-age dynamics in the Colombian Andes: developing an understanding of our legacy. Philosophical Transactions of the Royal Society London B 359, 173-181.

Hooghiemstra, H., Agwu, C.O.C., Beug, H.-J., 1986. Pollen and spore distribution in recent marine sediments: a record of NW-African seasonal wind patterns and vegetation belts. Meteor' Forschungs Ergebnisse, Reihe C 40, 87-135.

Hooghiemstra, H., Lézine, A.-M., Leroy, S.A.G., Dupont, L., Marret, F., 2006. Late Quaternary palynology in marine sediments: a synthesis of the understanding of pollen distribution patterns in the NW African setting. Quaternary International 148, 29-44.

Hoorn, C., Wesselingh, F.P., ter Steege, H., Bermudez, M.A., Mora, A., Sevink, J., Sanmartín, I., Sanchez-Meseguer, A., Anderson, C.L., Figueiredo, J.P., Jaramillo, J., Riff, D., Negri, F.R., Hooghiemstra, H., Lundberg, J., Stadler, T., Särkinen, T., Antonelli, A., 2010. Amazonia through time: Andean uplift, climate change, landscape evolution, and biodiversity. Science 330, 927-931.

Marchant, R., Behling, H., Berrio, J.C., Cleef, A., Duivenvoorden, J., Hooghiemstra, H., Kuhry, P., Melief, B., Van Geel, B, Van der Hammen, T., Van Reenen, G., Wille, M., 2001. Mid- to late-Holocene pollen-based biome reconstructions for Colombia. Quaternary Science Reviews 20, 1289-1308.

Torres, V., Hooghiemstra, H., Lourens, L.J., Tzedakis, P.C. 2013. Astronomical tuning of long pollen records reveals the dynamic history of montane biomes and lake levels in the tropical high Andes during the Quaternary. Quaternary Science Reviews 63, 59-72.

Van der Hammen, T. & González, E. (1964). A pollen diagram from th Quaternary of the Sabana de Bogota (Colombia) and its significance for the geology of the northern Andes. Geologie en Mijnbouw 43, 113-117.

Van der Hammen, T., Werner, J.H. & Van Dommelen, H., 1973. Palynological record of the upheaval of the Northern Andes; a study of the Pliocene and Lower Quaternary of the Colombian Eastern Cordillera and the early evolution of its high-Andean biota. Palaeogeography Palaeoclimatology Palaeoecology 16, 1-122.

Van 't Veer, R., Ran, E.T.H., Mommersteeg, H.J.P.M., Hooghiemstra, H., 1995. Multivariate analysis of the middle and late Pleistocene Funza pollen record of Colombia. Mededelingen Rijks Geologische Dienst 52, 195-212.

Wijninga, V.M., 1996. Paleobotany and palynology of Neogene sediments from the high plain of Bogotá (Colombia): evolution of the Andean flora from a paleoecological perspective. PhD thesis, University of Amsterdam, The Netherlands, 370 pp.

Henry Hooghiemstra, Suzette Flantua

University of Amsterdam

November 2017

Clim. Past Discuss.,
https://doi.org/10.5194/cp-2017-129-AC5, 2018

[Figure]

RESPONSE We thank Flantua and Hooghiemstra for the background information on scientific history.

FLANTUA Specific comments in text:

line 15: The pollen record presented in this paper reflects the interval 4.7-4.2 Ma only in 46 pollen samples. There is no justification why not a longer interval has been analysed.

RESPONSE The period studied in detail covers half a million years, which should be enough to cover the climatic variability. The window was chosen to cover the period of closure of the CAS as indicated by divergence of salinity of the surface waters between the Caribbean and the Eastern Pacific and highlighted in Fig. 1 (Figure 1 and Introduction lines 35-39).

FLANTUA line 18: The sediment core is located at the equator and one may wonder which area is considered as 'study area'. Fig. 2b offers the possibility to better show the potential pollen transport routes from the various source areas to site ODP 1239. We guess that pollen source areas stretch more to the south than to the north.

RESPONSE We'll change Figure 2 to better illustrate land-sea connections and the pathways of pollen transport (see separate file AC2). We consider western Ecuador, northernmost Peru and southwestern Colombia the main source areas of pollen and spores in sediments of ODP Site 1239. We'll add to Section 1.1.2: the following two paragraphs replacing lines 139-143):

"Ríncon-Martínez et al. (2010) showed that the terrigenous sediment supply at ODP Site 1239 during Pleistocene interglacials is mainly fluvial and input of terrestrial material drop to low amounts during the drier glacial stages. In addition, transport of pollen and spores to the ocean is mainly fluvial (González et al., 2010). High rates of orographic precipitation characterize the western part of equatorial South America. These heavy rains quickly wash out any pollen that might be in the air and result in large discharge by the Ecuadoran Rivers (Fig. 2). Esmeraldas and Santiago Rivers mainly drain the northern coastal plain of Ecuador, and the southern coastal plain is drained by several smaller rivers, which end in the Guayas River. Moreover, the predominantly westerly winds (Fig. 2) are not favorable for eolian pollen dispersal to the ocean. Nevertheless, some transport by SE trade winds is possible and should be taken into account.

"After reaching the ocean pollen and spores might pass the Peru-Chile Trench; which is quite narrow along the Carnegie Ridge, by means of nepheloid layers at subsurface depths. Some northward transport from the Bay of Guayaquil by the Coastal Current (Fig. 2) is likely. However, the Peru-Chile Current flows too far from the coast to have strong influence on pollen and spore dispersal. We consider western Ecuador, northernmost Peru and southwestern Colombia the main source areas of pollen and spores in sediments of ODP Site 1239."

FLANTUA l.30: Montes 2015 should be Montes et al. 2015.

RESPONSE We'll correct Montes 2015 into Montes et al. 2015, also in the reference list.

FLANTUA If you really want to capture the discussion around the closure, then you should already add here: O'Dea et al. 2016a. Formation of the Isthmus of Panama. Science Advances. O'Dea et al. 2016b Building bridges. Response to Erkens and Hoorn: "The Panama Isthmus, 'old', 'young' or both?" Science Advances

RESPONSE O'Dea et al. (2016) has been cited on line 31.

FLANTUA l.33. See also figure 1 from O'Dea et al. 2016b

RESPONSE We prefer our Figure 1 focusing on the surface water exchange between Caribbean and Eastern Pacific over the schematic figure of O'Dea et al. 2016b, because we'll later focus on changes in hydrology. We avoid a discussion about biogeography and faunal exchange between the Americas because this is beyond the scope of the paper.

FLANTUA l.57: 'Northern Andes' needs to be better specified as the uplift history in Ecuador differs from Colombia.

RESPONSE The paragraph concerns aspects of the uplift of both the Ecuadorian Andes, the Central Andes, and the Eastern Cordillera of Colombia. We thus describe the uplift of the Ecuadorian Andes in a wider context, while specific remarks about the Andes of Ecuador are given at lines 61, 104 and 363. We'll specify Ecuadorian Andes at lines 357, 425 and 445.

FLANTUA l.63: Additionally to Spikings et al. 2005: see Supplementary Information

Fig. S1 in Hoorn et al., 2010.

RESPONSE Hoorn et al. (2010) reviews Spikings et al. (2005). We prefer the primary reference.

FLANTUA l.74: first palynological record = first marine palynological record. Pollen records of Pliocene age are Rio Frio-17 and Subachoque-39 (Wijninga, 1996 figures 8.2 and 8.4;Hooghiemstra et al., 2006 figures 4 and 7) that show the floral composition of montane forest and páramo.

RESPONSE We'll add a reference to the work of Wijninga and alter the sentence at lines 73-75 to: "While other palynological studies of the region have been conducted for the mid-Pliocene to Holocene (González et al., 2006; Hooghiemstra, 1984; Seillès et al., 2016), only a few palynological records for the early Pliocene exist (Wijninga and Kuhry, 1990; Wijninga, 1996)."

FLANTUA Table 4 of the 2006-paper may be helpful in considering how pollen taxa are grouped to reflect the altitudinal vegetation zones.

RESPONSE We do not understand how Table 4 of Hooghiemstra et al. (2006) could help in the classification of elements like Polylepis/Acaena. No specific pollen types for the páramo vegetation are listed in this table (only the broad range families Poaceae, Ericaceae and Asteraceae).

FLANTUA l.81: make a difference between the 'Western Cordillera' in Ecuador and the 'Western Cordillera' in Colombia (see also lines 62-63).

RESPONSE We'll change "Western Andean Cordillera of western equatorial South America" into "western Andean Cordillera of Ecuador"

FLANTUA l.85: Figure 2b shows 'terrestrial biomes' in 7 legend units. However, this palynological study is using 4 legend units (the left hand four in Table 1). The link between pollen source areas and pollen diagram can be improved if for consistency Figure 2b would show the same categories as recognized in Table 1. Fig. 2b can be improved by indicating with arrows the routes and mechanisms (ocean currents, rivers, wind?) of pollen and spore transport towards ODP 1239. The information of the inset figure can be placed (with less detail) in the main figure.

RESPONSE We'll change Figure 2 to better show the connection between rainfall in northern South America and the latitudinal position of the ITCZ and illustrate the transport mechanisms of wind and river to the ocean (see new Figure 2 in the supplementary file AC2). We'll also add the 1000, 2300, 3200, and 4800 m contours that are used in defining LR, LMF, UMF, and Páramo, respectively (see new Figure 2, supplementary file AC2).

FLANTUA l.95: The acronym ENSO has been introduced already in line 49.

RESPONSE We'll check all acronyms for necessity and consistency.

FLANTUA l.96: Marchant et al., (2001) is a good example of small-scale climate patterns.

RESPONSE We'll add "(e.g. Marchant et al. 2001, Niemann et al. 2010)"

FLANTUA l.118: Here you specify "the mountains" as the area within Ecuador, while before you talk broadly about the "Northern Andes". For consistency, the Ecuadorean Andes/Cordilleras should be used throughout the paper.

RESPONSE We'll specify Ecuadorean Andes where appropriate; that is at lines 22, 357 and 425. We'll also check the consistency of the capitalisations.

FLANTUA l.127: Palmae = Arecaceae

RESPONSE We'll change Palmae to Arecaceae.

FLANTUA l.141: Santiago river not shown in map?

RESPONSE We'll add the Santiago River on the vegetation map.

FLANTUA l.140-141 & 162: The combination of pollen and fern spores in the pollen sum is not ideal. In Lowland Forest (< 1000 m) and Lower Montane Forest fern-loaded trees fallen across small rivers shed enormous amounts of spores directly into the water currents (personal observations). Therefore, the proportion of fern spores in the pollen spectra are of a different character than the proportion of pollen, and both components are difficult to compare (see also figures 72-75 in Hooghiemstra et al., 1986).

RESPONSE It makes sense to add the spores to the pollen sum as both are terrestrial derived. The maps of Hooghiemstra et al. 1986 show a clear relation to the distribution of the rain forest on the African continent and the spore abundances in the marine sediments. Dupont & Agwu (1991) showed that the distribution of monolete spores in modern marine sediments compares well with that of Elaeis guineensis (Dupont, L.M. and Agwu, C.O.C., 1991. Environmental control of pollen grain distribution patterns in the Gulf of Guinea and offshore NW-Africa. Geologische Rundschau, 80: 567-589.) In some marine pollen diagrams (though not in the present study), it is advisable to leave out pollen from typical coastal habitats such as mangroves (Dupont & Weinelt , 1996: Vegetation history of the savannah corridor between the Guinean and the Congolian rain forest during the last 150,000 years. Vegetation History and Archaobotany, 5: 273-292.)

FLANTUA l.143: It would be very useful to have figure 2 show these geographical features such as the Carnegie Ridge and Peru-Chile Trench, and show how pollen would reach the location of the record.153-154: What is the potential pollen supply by the Humbold Current ? and where could pollen source areas be located?

RESPONSE We'll add bathymetry to vegetation map. Much pollen probably reaches the Carnegie Ridge by means of nepheloid layer transport after reaching the ocean by river discharge. Studies of late Quaternary terrestrial input clearly show the importance of fluvial discharge (González et al., 2006; Ríncon-Martínez et al., 2010). Dominant winds are not favorable for pollen transport to the marine site. The Humboldt Current does not reach the area (we correct that in the new Figure 2, see supplementary file

AC2) and even the Peru-Chile Current flows too far from the coast to be considered an important transport mechanism of pollen grains to our site. Some transport over relative short distances by the Coastal Current (new Figure 2 in supplementary file AC2) is to be expected. We'll add these remarks to the section and the caption of the new Figure 2 (see above, response to line 18).

FLANTUA Furthermore, from this description it's not clear why you are expecting to find signals from the eastern lowlands which constitute the western margin of the Amazon Basin (line 118).

RESPONSE We'll correct the confusing formulation and start the sentence at line 117 with "Ecuador is geographically...." and the next sentence with: "North of Ecuador, the mountains. . .."

FLANTUA l.156: Avoid starting the sentence with a number. See 191.

RESPONSE The beginning of the paragraph changes anyhow because of the extra samples included: "A total of 68 samples of 10 cm3 volume have been analyzed. For the interval between 301 and 334 m (4.7 and 4.2 Ma),"

FLANTUA l.183 / Table 1: (a) Table 1 shows an effort to group pollen and spore taxa into meaningful ecological groups. A marine pollen record has a wide pollen source area and many taxa listed do have a wide ecological range. The latter makes it difficult to develop clear-cut ecological groups. Páramo: Several 'páramo' taxa also occur in the forest; Upper Montane Forest: Melastomataceae from the UMF also occur abundantly in the páramo; Lowland Forest (LF): is represented by a remarkably low number of taxa; Broad-range taxa: Poaceae indeed have a broad ecological range, but most of it comes from páramo and also dry vegetation in coastal areas and interandean valleys; Asteraceae Tubuliflorae also have a wide range but the bulk is from the páramo; Artemisia seems from Peruvian origin?; Thevetia is indicator of dry conditions rather than a broad range plant; Humid indicators: the following 4 taxa are advised to be omitted from this list: Ranunculaceae (important plant in the puna), Ilex (indeed, also in wet forest but frequently elsewhere), Myrica (= Morella), and Malpighiaceae (also in savannas).

RESPONSE Indeed, these groupings cannot be 100% sure and some subjectivity is unavoidable. We do not want to include pollen of large families such as Poaceae, Ericaceae, and Asteraceae into the Páramo group, because we try to leave that group as exclusive as possible. If we had included these pollen types into the Páramo group, critics concerning the lack of conclusiveness would have been justified. The group of Humid Indicators is dominated by Cyperaceae pollen and fern spores; thus, the changes you propose have very little impact on the percentage values of the group.

FLANTUA (b) A significant proportion of Lowland Forest (including wet rainforest and dry deciduous forest) and Lower Montane Forest is palynologically 'silent' as many trees are pollinated by insects, beetles, etc. Given the northward flow of the Humboldt Current, and atmospheric circulation in the direction of the coast, I am not surprised that LF is hardly represented by taxa in Table 1. Palms are more abundant in Lower Montane forest (LMF); by moving the Arecaceae from the category LF to LMF, LF is hardly reflected at all. This has consequences for the interpretation and conclusions.

RESPONSE We agree that the representation of the lowland rainforest is weak and is dominated by Polypodiaceae spores. Arecaceae pollen are Wettinia-type form the low-land rainforest (we'll correct the wrong naming in the original supplementary Figure S1) and Socratea distributed in the lowland rainforest from Nicaragua to Bolivia (Marchant et al., 2002). We'll specify the two Arecaceae Wettinia and Socratea in Table 1. We combine Figs. S1-S3 into 1 supplementary figure (see supplementary file AC2). As already stated before, the Humboldt Current does not reach the Carnegie Ridge, but the northbound Coastal Current might have lessened the representation of the lowland forest.

FLANTUA (c) Most of the taxa in the category 'páramo' occur in the ecotone zone of the UMF. Thus the record for 'paramo' as based on the taxa listed in Table 1 may also reflect the zone with dwarf forest and shrub. (d) In Flantua et al. (2014, figure 7) we explored how altitudinal vegetation zonation is reflected on the basis of GBIF data. We were quite disappointed with the poor altitudinal zonation, possibly explained by a large amount of 'noise' in the GBIF data by using data from a wide geographical region. We concluded that up to date an altitudinal zonation based on expert knowledge from field botanists gives better results: see Groot et al., 2013 (RPP) figure 3.

RESPONSE We group very few taxa in the Páramo and leave out pollen of large families such as Poaceae, Ericaceae and Asteraceae, which other authors (e.g. Hooghiemstra et al., 2006) do include in the Páramo group. Thus, we are very cautious at this point. One of the most important pollen types is Polylepis/Acaena (Figure 5) included in the subpáramo by Flantua et al. (2014), which treat sub-, grass-, and superpáramo as one unit. They mention on page 112 of their paper: "The distribution of subpáramo species does not differ from the grasspáramo species; both the nuclei and edges show similar patterns. This justifies the strategy chosen for this analysis, to assess páramo dynamics as a single biome." Therefore, we are convinced that we selected proper indicators for Páramo. The curve of Asteraceae pollen in Figure 5 is only given for comparison. We do not use the Asteraceae record as an indicarion of Páramo.

FLANTUA l.185: Here, make also reference to Van der Hammen (1974).

RESPONSE We suppose you mean Van der Hammen et al. 1973 (as mentioned in your reference list).

FLANTUA l.188: Which taxa occurred in less than 10% ? A Table showing all identified pollen and spore taxa and their assessment to ecological categories would be helpful.

RESPONSE Table 1 does list ALL taxa as stated in the heading "List of identified pollen and spore taxa…." We regret that the figures in the supplement are badly readable. We will amend this in with a new supplementary figure combining Figs S1-S3 (see supplementary file AC2).

FLANTUA l.192: to what refers 'respectively'?

RESPONSE To pollen concentration of 685 grains/cm3 and spore concentration of 465 grains/cm3.

FLANTUA l.194: 26-27% ? Does this relate to a figure?

RESPONSE: Figure 3

FLANTUA l.197: Where is the replacement of Podocarpus by Alnus shown in a figure?

RESPONSE We add values for Podocarpus to Figure 3. We'll replace "whereof the.. Alnus" (line 197) with "During the Holocene Podocarpus is replaced by Alnus as the most abundant upper montane forest tree, although Podocarpus was still abundant during the glacial (González et al. 2010)."

FLANTUA l.207: 'The Lowland Rainforest is mainly represented by Polypodiaceae'. However, Polypodiaceae actually occur everywhere and are not a representative of lowland rainforest in particular.

RESPONSE Yes, that is true. However, Polypodium is a most common epiphytic on lowland trees (Marchant et al. 2002)

FLANTUA Figure 4: curves are easier to read with a horizontal line at 0%.

RESPONSE We alter the figure and add baselines (see supplementary file AC2).

FLANTUA The elevation indicated for páramo and UFL overlap by 100 m. Is this on purpose?

RESPONSE No, this is a mistake, which we'll correct. We'll take the 3200 meter contour for the transition between Páramo and UMF.

FLANTUA Can your modern samples be shown here as well for comparison?

RESPONSE We make an extra figure (temporally called Fig 4A) comprising samples from the Pleistocene and Holocene. (see Figure 4A in supplementary file AC2)

FLANTUA l.217-237: make reference to the figure where all these changes can be seen.

RESPONSE We'll add "(Fig. 4 and Supplementary Figure)" at line 217.

FLANTUA l.229: 'of the pollen sum' is redundant.

RESPONSE: OK

FLANTUA l.232-233: Podocarpus excluded from the UMF; if I understand well, not from the pollen sum.

RESPONSE: Yes

FLANTUA l.240: 2 out of the 3 species mentioned also occur in the forest (Polylepis, Huperzia). The species mentioned are not a strong indication of the presence of páramo. Lycopodium with foveolate spores is most characteristic of páramo vegetation (Van 't Veer et al., 1992); absence of Lycopodium fov. in the pollen spectra is in support of the view that the present taxa identified as 'reflecting páramo' also are reflecting lowermost páramo and ecotone forest.

RESPONSE We strongly disagree. Your observation that Lycopodium fov would be absent in our record is wrong. Huperzia spores are foveolate (Rincón Baron et al., 2014: Esporogénesis, esporodermo y ornamentación de esporas maduras en Lycopodiaceae in Rev. Biol. Trop., 62: 1161-1195). If you want you may read Huperzia as 'Lycopodium with foveolate spores'. As mentioned above Polylepis is grouped in the subpáramo by yourself (Flantua et al., 2014). Moreover, you also mention: "the distribution of subpáramo species does not differ from the grasspáramo species". Together, with the presence of Jamesonia/Eriosorus spores our argument is valid that Páramo vegetation existed in the Ecuadorian Andes (and maybe beyond) for at least 6 Ma.

FLANTUA l.245: Fig. 5: where is the curve showing páramo vs. montane forest?

RESPONSE: Figure 4 & Figure 7

FLANTUA 'páramo sum' is a confusing term; better to use 'paramo taxa (%)'

RESPONSE OK, we leave out 'sum'. We also changed Fig 5 to show the trends over the full Pliocene and Pleistocene.

FLANTUA l.251-253: difficult to understand why the evidence mentioned is suggesting drier conditions. Please explain more clearly.

RESPONSE Yes, this is not very clear. We'll delete the sentence "This together…drier conditions" (line252-253) and rewrite section 4.1 as follows: "In order to better understand the source areas and transport ways of pollen grains to the sediments, we make a comparison of the results of our two Holocene samples with that of another pollen record retrieved from the Carnegie Ridge southeast of Site 1239 (Figure 2) reflecting rainfall and humidity variation of the late Pleistocene (González et al. 2006). Holocene samples of Site 1239 gave similar results showing extensive open vegetation (indicated by pollen of Poaceae, Cyperaceae, Asteraceae) and maximum relative abundance of fern spores although concentration is low González et al., 2006). As also indicated by the elemental ratios, fluvial transport of pollen predominates in this area (González et al., 2006; Ríncon-Martínez, 2013). This is understandable as both ocean currents and wind field do not favor transport from Ecuador to Site 1239 (Figure 2)."

FLANTUA l.255: better to show the location of TR 163-38 on the map in Fig. 2.

RESPONSE Yes, we show the position in the new Figure 2 (see supplementary file AC2).

FLANTUA l.267: difficult to understand the claim '(below the forest line)'. How can open grassland below and above the UFL be identified and separated from each other? This seems an over-interpretation of the data.

RESPONSE Apparently, our phrasing has been misunderstood. We'll therefore change "(below the forest line)" to "(apart from Páramo)"

FLANTUA l.270: 'expansion of savannas' ? whereas (line 266) there is a near absence of open grasslands.

RESPONSE Here, we cite Salzmann et al. (2011) and refer to global vegetation change. To amend the confusion, we'll add "in Africa, for instance"

FLANTUA l.272: 'All altitudinal vegetation belts are already present'. Most possibly correct but not necessarily in its present form. For example, after Quercus had changed the composition of montane forest (LMF and UMF) several LMF taxa were able to reach higher elevations (Hooghiemstra, 1984; Torres et al., 2013). Unfortunately, modern climatological constraints of the lower and upper boundaries of the LMF are insufficiently understood (Hooghiemstra et al., 2012); as a consequence it is difficult to infer climatological change from altitudinal migrations of LMF.

RESPONSE Therefore, we cautiously only remark that "All altitudinal vegetation belts are already present, with varying ratios,"

FLANTUA l.272: 'goes through the most prominent' is unclear, please rephrase.

RESPONSE We'll change the lines 272-273 to: "belts are already present, with varying ratios, and only pollen percentages of lowland rainforest rise from almost absent to 6%."

FLANTUA l.278: 'It is known from other Andean pollen records ....' The comparison made here should be better explained.

l.277: 'show a similar pattern of expansion': what do you mean exactly?, and where can the reader see this expansion?

l.279-282: unclear text, needs rephrasing. For instance, unclear use of 'opposing', what pattern is exactly to be seen in the fig. 5, what is the "more general pattern"?

RESPONSE We'll rephrase lines 274-282 to: "Shifts in the vegetation are driven by various parameters such as temperature, precipitation, CO2, radiation, and any combination thereof. However, a hint to which parameter has strongest influence on the vegetation might be given by the pattern of expansion and retreat of different vegetation belts.

Hooghiemstra and Ran (1994) indicate that if temperature were the dominant driver of vegetation change, altitudinal shifting of vegetation belts would lead to increase in the representation of one at the cost of another. We do not see such a pattern in our record with the possible exception in zone III where the trends between pollen percentages of Páramo and those of upper montane forest (without Podocarpaceae) are reversed. However, the more general pattern indicates parallel changes in the representation of the forest belts suggesting that not temperature but humidity had the stronger effect on the Pliocene vegetation of Ecuador."

FLANTUA l.287 'the main transport agent for pollen' ; I guess also for spores? Replace here and in the following sentences "El Niño" by ENSO.

RESPONSE In this case, we discuss the warm phase of ENSO that is called El Niño and thus it is better to use that terminology; ENSO comprises all three phases of the oscillation. We'll add: ", the warm phase of ENSO" after El Niño on line 287.

FLANTUA l.292: it would be useful to have a figure that can support this statement on 'lowland rainforest of the coastal plain further north' as it's unclear what "further north" is. Or indicate with lat/long values.

RESPONSE We'll change "further north" into "of Ecuador and western Colombia"

FLANTUA l.294: 'lowland rainforest' is poorly reflected by the taxa listed in Table 1. and as a consequence it is difficult to make a comparison.

RESPONSE We add an extra figure showing the long-term trend of Pliocene and Pleistocene (see Fig. 4A in supplementary file AC2). We'll delete the last sentence of the paragraph.

FLANTUA l.298: Difficult to understand what means 'the development of pollen values is decoupled from'. Needs a better explanation and visualization.

l.300 'eolian' transport is contra to line 140-141.

RESPONSE We'll rephrase lines 298-299: "The trend in pollen percentages of Podocarpaceae divert from that of the other pollen taxa, which may be explained by additional transport of Podocarpaceae pollen by wind. The high pollen production of "

We'll also changed lines 139-143 (see above, response to the comment on line 18).

We'll change lines 305-306 to: "of the easterly trade winds. Increase in trade wind strength at 4.4 Ma would be in line with a shift in the locus of maximum opal accumulation rates in the ocean associated with a shift in nutrient availability (Farrell et al., 1995). Dynamic modelling indicates that stronger easterlies would cause shoaling of the EEP thermocline (Zhang et al., 2012), which took place between 4.8 and 4.0 Ma (Steph et al.," and delete "Another noteworthy oceanographic change occurred at 4.4 Ma in the EEP. Farrell et al. (1995) described a shift in the locus of maximum opal accumulation rates from ODP Site 850 to ODP Site 846 (Galápagos region), caused by a shift in the availability of nutrients, which is possibly related to increased trade wind strength after 4.4 Ma." (lines 314-316)

FLANTUA l.317: 'Besides being influenced by hydrological changes and wind strength' is unclear and needs further explanation.

RESPONSE We'll rephrase lines 317-322 as follows: "Comparing the pollen percentages of Páramo and upper montane forest (Fig. 7) indicates that UMF maxima coincide with Páramo minima and SST maxima at ODP Site 846 (Lawrence et al., 2006). This might be explained by a shift of the upper montane forest to higher altitudes at the cost of the area occupied by Páramo vegetation as a result of higher atmospheric temperatures or increased orographic precipitation in the western Andean Cordillera caused by higher SST and increased evaporation."

FLANTUA l.321: Replace western Andean Cordillera with western Cordillera of Ecuador.

RESPONSE Increase in precipitation would not be restricted to the Cordillera of

Ecuador.

FLANTUA Be consistent throughout the text with Western or western.

RESPONSE We'll stick to "western"

FLANTUA l.324: Sums of upper montane forest = Representation (%) of upper montane forest

RESPONSE We'll change it to "Pollen percentages of"

FLANTUA l.334: better to use the more recent reference '(Hoorn et al., 2010)'

RESPONSE We'll add the reference of Hoorn et al. 2010.

FLANTUA l.333+335: Add 'Ecuador' to Eastern and Western Andean Cordillera

RESPONSE No, the process described would have influenced the Colombian Andes as well.

FLANTUA l.342: In order to use páramo vegetation = In order to use the abundance of páramo vegetation

RESPONSE We'll add "the existence of"

FLANTUA l.343: Replace 'no true páramo endemics' by 'Although no taxa restricted to páramo only were identified. . ..'

RESPONSE: Done

FLANTUA l.347: 'Polylepis is reaching 5000 m in the northern Andes': I guess this refers to Peru and Bolivia and maximum elevations relate to individual trees. In Colombia and Ecuador Polylepis dwarf forest occurs up to 4200-4300 m.

RESPONSE The citation is incorrect. On line 347 is written: "which forms the transition to other forest types and up to 5000 m (Kessler, 2002)." We'll add "in Bolivia"

FLANTUA l.349-351: Perhaps not as present in montane forest and lowland rainforest, but relatively close to your marine record, you have the presence of several major forest nuclei of seasonally dry tropical forest biome (see Särkinen et al. 2011) and there are a number of different species of Asteraceae in Peruvian seasonally dry tropical forest (see book Neotropical Savannas and Seasonally Dry Forests: Plant Diversity, Biogeography, and Conservation by T. Pennington & J. Ratter 2006). Could this biome be the source of Asteraceae in your record?

RESPONSE That possibility is exactly why we do not use Asteraceae as an indicator of Páramo.

FLANTUA l.354: 'without changes in composition' is rather meaningless as so few páramo taxa have been identified.

RESPONSE We'll delete "without changes in composition"

FLANTUA l.355: which evidence is fueling this assumption?

RESPONSE The reasoning that mountainous vegetation could grow at higher altitudes during warmer periods.

FLANTUA l.356-357: the weak evidence of páramo does not allow to infer conclusions about the elevation of the Andes.

RESPONSE Our evidence of Páramo is not weak. We show a record of continuous presence since the early Pliocene of pollen of plants (Polylepis, Huperzia, Jamesonia) that grow in subpáramo and páramo and that do not grow at lower altitudes (see also the Figure 4A in supplementary file AC2).

FLANTUA l.362-368: uplift histories of the various areas are confusing here:

RESPONSE Nevertheless, it is necessary to mention the different published opinions. We'll delete "uplift history for the western Cordillera of the Northern Andes and according"

FLANTUA l.362: indeed uplift is older as can be seen in Hoorn et al., 2010. l.364: uplift of the Central Andes is 60-25 Ma (instead of 10-6 Ma; see Hoorn et al., 2010, Suppl. Info. )

RESPONSE We speak about the major uplift phase.

FLANTUA l.365: Amazon fan = Amazon Fan

RESPONSE: OK

FLANTUA l.365: Which is the first palynological paper to state here "in another recent palynological study.."?

RESPONSE Hoorn et al. (2017) cited at the end of the sentence (as usual).

FLANTUA l.366: The Hoorn et al. 2017 paper suggests but does not provide conclusive evidence that the grass pollen are from páramo as the source area for the Amazon river include also high Andean open vegetation of the puna. This sentence here should be rephrased to not 'oversell' Hoorn et al. 2017 in support of páramo presence.

RESPONSE Hoorn et al. (2017) identified Polylepis and Huperzia (Lycopodium fov) in sediments of the Amazon Fan. Poaceae and Asteraceae are listed by Hoorn et al (2017) as widely distributed and not included in the group of Páramo indicators. Neither Hoorn et al. nor we claim that grass pollen exclusively comes from the Páramo.

FLANTUA l.377: Amaranthaceae and Thevetia rather are reflecting dry conditions.

RESPONSE Indeed. However, Thevetia occurred with a few exceptions in the earliest Pliocene samples only (older than 4.7 Ma).

FLANTUA l.379: what is the meaning of 'all altitudinal vegetation belts go through simultaneous shifts of expansion and retreat' ?

RESPONSE We'll delete this confusing sentence.

FLANTUA l.382: Add space before the 3.

RESPONSE: OK

FLANTUA l.385: Better explain 'parallel expansion and retreat of all vegetation belts'. For the last 20 ka we have learned that little goes parallel (see Hooghiemstra and Van der Hammen, 2004).

RESPONSE We'll rephrase the sentence as follows "Our record does not show increased representation of one vegetation belt at the cost of another indicating that altitudinal shifts were not extensive and moisture availability might have been an important driver of Pliocene vegetation change." and start the next sentence with "Changes in humidity could..."

FLANTUA l.419: Eastern Cordillera reached = Eastern Cordillera of Colombia reached

RESPONSE: Done

FLANTUA l.421: 'argue for a rapid rise of the region since 4-6 Ma' ; This is outdated and should be 30-5 Ma (see Hoorn et al., 2010 Suppl. Info.)

RESPONSE Indeed. We'll change "argue" into "argued"

FLANTUA l.425: 'Our pollen record from the páramo shows .....' This conclusion seems unwarranted as the evidence for páramo vegetation is weak and also could reflect ecotone forest and/or other biomes.

RESPONSE We can only repeat that we use very specific Páramo indicators and do NOT use the record of broad range taxa such as Asteraceae and Poaceae.

FLANTUA l.435: On which evidence is this sentence based?

RESPONSE This has been extensively discussed in the previous section.

FLANTUA l.440: Conclusion 2 is difficult to understand: when? a shift to what?

RESPONSE We'll rephrase as follows: "The most prominent shift recorded is an increase in the representation of the lowland rainforest."

FLANTUA l.441: Higher representation of Podocarpaceae is interpreted as evidence of more intense trade winds. However, this is not necessarily the case as pollen record Funza09 (Torres et al., 2013, figure 10) shows that Podocarpus is more abundant during several intervals of Pleistocene time, potentially also leading to high representation in the marine sediments.

RESPONSE Our argumentation is not based on the abundance of Podocarpaceae pollen but on the divergence between the trends in pollen percentages of Podocarpaceae and those of other pollen and spores (see Section 4.2.2)

FLANTUA l.447-448: The presence of páramo is weakly supported by evidence; the inferred altitude of the Ecuadorian (?) Andes is speculative as a consequence

RESPONSE We can only repeat that we use very specific Páramo indicators and do NOT use the record of broad range taxa such as Asteraceae and Poaceae. . FLANTUA l.449-450: Better to refer to more recent literature in which the uplift of the Northern Andes has been set back in time already.

RESPONSE We'll rephrase: "We present new paleobotanical evidence indicating an earlier development of Páramo vegetation than . . ."

FLANTUA l.564: Reference Montes et al. 2015 is incomplete.

RESPONSE We'll correct the reference of Montes and also the missing parts of the references of Ríncon-Martínez 2013, Sanchez-Baracaldo, 2004 (American Fern Journal 94(3):126-142. 2004), Flantua et al. 2014.

FLANTUA Fig. S1: To which degree modern core top samples are comparable to the pre-Quaternary samples? Are mechanisms of pollen transport comparable? Some re-marks about this issue are missing.

RESPONSE All transport mechanisms were already in place during the Pliocene. However, some enhancement of SE trade winds might have been captured by the Podocarpus pollen record, which is discussed in Section 4.2.2.

FLANTUA Fig. S2: % sum páramo = páramo (place the word 'percentage' in the figure caption) :also for other taxa

RESPONSE: OK

FLANTUA Fig. S3: Mention in the caption 'Pollen percentage diagram' and omit all % % indications on top of the pollen diagram. And: Myrica = Morella

RESPONSE Figure S3 will be replaced (see supplementary figure in supplementary file AC2). Myrica is corrected to Morella

FLANTUA IN CONCLUSION: The biomes 'páramo' and 'lowland rainforest' are hardly reflected by characteristic pollen and spore taxa. Several taxa now classified as 'broad range taxa' cold be shifted to 'páramo' but with the same restriction that these taxa also could reflect uppermost montane forest (ecotone forest).

RESPONSE First you tell us to include Poaceae, Asteraceae, and Ericaceae in the Páramo group and subsequently you argue that these families are too widely distributed to indicate Páramo vegetation. We deliberately did not use broad-range families as indicator for Páramo, but used specific Páramo indicators such as Polylepis-Acaena pollen and spores of Lycopodium fov (Huperzia) and Jamesonia/Eriosorus. We thus have a strong record of early Pliocene Páramo vegetation in the Ecuadorian Andes. You try to refute our claim by erroneously stating or implying that our Páramo indicators would be based on broad range families such as Asteraceae, Ericaceae and Poaceae.

FLANTUA In marine pollen records changing proportions of pollen taxa / ecological groups may reflect vegetation change and / or changes in pollen transport. In the present manuscript the latter is hardly/not considered (This remark also relates to the suggestions for improvement of Fig. 2).

RESPONSE We'll expand the section about transport source areas of pollen and spores (Section 1.1.2., see response to comment on line 18). We consider the ef- fects of transport mechanisms as we discuss different transport mechanisms in the case of the Podocarpus record (Section 4.2.2). Fig. 2 has been changed to better illustrate the possible transport routes (see supplementary file AC2).

FLANTUA Integration of terrestrial and marine proxies is a powerful tool to maximize conclusions. The comparison with model output has broadened the scope of this paper but – apart from speculation - has not generated an incremental step forwards.

RESPONSE Based on our data, we clearly take position about southward shift of the Pliocene ITCZ pro data and contra modelling results (Lines 380-400) and discuss several aspects of the problem (Lines 401-428). We protest against the disqualifying phrase "apart from speculation".

FLANTUA Pollen zones in Fig. S3 are not expressive and the interpretation in terms of environmental change is not convincing. The presented pollen evidence does not allow a full support of the suggested conclusions of this paper. Analysing a much longer interval has the potential to strengthen conclusions, but the regional setting will remain poor to obtain convincing evidence.

RESPONSE We agree that Figs S1-S3 are barely readable and we replace them with a new supplementary figure combining Fig. S1, S2 and S3 (see supplementary file AC2). The pollen zones are supported by the cluster analysis (CONISS), which is shown in the first panel of the supplementary figure. We originally used the coring gap as a zone boundary, which we agree is not correct. Therefore, we now only use the CONISS clustering to define the pollen zones. This results in the extension of pollen zone II downward (see supplementary figure in file AC2). We'll adapt the description and further carry out all necessary corrections. However, the shift in the boundary between pollen zones I and II does not affect the discussion or the conclusions. We make an extra figure (see Fig 4A in supplementary file AC2) showing at low resolution the trends in Pliocene and Pleistocene.

A detailed study of the upper Pliocene is in preparation.

ADDITIONAL REFERENCES

Marchant, R., Behling, H., Berrio, J.C., Cleef, A., Duivenvoorden, J., Hooghiemstra, H., Kuhry, P., Melief, B., Van Geel, B., Van der Hammen, Th., Van Reenen, G., Wille, M., 2001. Mid- to Late-Holocene pollen-based biome reconstructions for Colombia. Quaternary Science Reviews, 20: 1289-1308.

Niemann, H., Brunschöm, C., Behling, H., 2010. Vegetation/modern pollen rain relationship along an altitudinal transect between 1920 and 3185 m a.s.l. in the Podocarpus National Park region, southeastern Ecuadorian Andes. Review of Palaeobotany and Palynology, 159: 69-80.

Wijninga, V.M., 1996. Pylanology and palaeobotany of the Early Pliocene section Río Frío 17 (Cordillera Oriental, Colombia): biostratigraphical and chronostratigraphical implications. Review of Palaeobotany and Palynology, 92: 329-350.

Wijninga, V.M. and Kuhry, P., 1990. A Pliocene Flora from the Subachoque Calley (Cordillera Oriental, Colombia). Review of Palaeobotany and Palynology, 62: 249-290.
Additional references:
* * *
Clim. Past Discuss.,
https://doi.org/10.5194/cp-2017-129-RC1, 2017

[Figure]

Paper: Early Pliocene vegetation and hydrology changes in western equatorial South America by Grimmer et al.

Reviewer Carina Hoorn

Summary The purpose of the paper is to establish the direction of shift of the ITCZ following the closure of the Central American Seaway (CAS) and uplift of the northern Andes. The paper comprises a palynological study of sediments from the interval between 4.7 and 4.2 Ma of the appropriately situated ODP core 1239A. The specific aims are to reconstruct vegetation, climate and topography in this region throughout this time interval. The conclusion is that an (already) high Andean landscape existed at the time, and that both vegetation and landscape during this interval match with a scenario corresponding to a southward shift of the ITCZ. Fluctuations of the ENSO are also considered. The results are in accordance with other paleoceanographic data in the region.

Main comments:

There is a shortage of continuous records from the Pliocene in the eastern Pacific that reflect hydrological and climatic change in the region. This paper aims to fill this gap. However, the dataset makes it hard to see the big changes that one would expect from the text. If possible the dataset should be extended with additional data to which are referred in the text.

• The interaction of Andean uplift, closure of the CAS, shifting ITCZ and ENSO altogether make it quite a daunting task to interpret the palynological diagram an assign changes to specific causes. The case is clearly made and looked at from all angles. Question: Is there a chance that some of the subtle changes in the diagram can be related to the Pliocene uplift pulses in the Andes and related atmospheric changes? Such pulses are postulated in tectonic reconstructions (e.g. Anderson et al., 2015, Geosphere) and are mentioned by authors in the paragraph starting at line 464.

• The new dataset further confirm that a high topography (Anderson et al., 2015) and paramo (Bermudez et al., 2015 in Basin Research; Hoorn et al., 2017 in Global & Planetary Change) was in place at least since the early Pliocene. It might be worthwhile highlighting the regional character of this condition?

Note that modern type precipitation patterns are likely to have been in place already from middle Miocene onwards (see Kaandorp et al., 2006; Hoorn et al., 2010; Barnes et al., 2012) and this would have required a significant orographic barrier. A high Andes might go as far back as the mid-Miocene, however, first evidence for a paramo is now set as latest Miocene to early Pliocene. Lines 406-407 could be reconsidered in this context.

• The elemental concentrations analysis needs to be better introduced and is currently rather hidden and makes a surprise first appearance in the methods section. In methods also explain why this is a useful additional technique. Part of the text in section 4.3 (line 360 onwards) could be moved to the introduction to explain approach.

• The discussion of the Holocene samples in relation to the Pliocene seems a bit ambivalent and does not form a very good guideline to better understand the new results.

• Lines from 313: A rather crucial line comes up here and reads as follows: "unpublished data from the earliest Pliocene show that the percentage of lowland rainforest before 4.7 Ma was very low". The evidence that is presented seems rather subtle and perhaps not iconic for an important vegetation & climate change. The authors allude to data of the earliest Pliocene, which they say strengthen their case. However, they are not visible. If these data belong to the authors it might be timely to include them here (or a selection of them) and make a more compelling case.

• A map with the scenarios for the changing ITCZ would be welcome. Instead this could also be added to figure 1.

Minor comments:

• In line 465 Hoorn et al. 2010 are listed as backing up a rapid rise of the region since 4– 6 Ma, However we suggest in the mid-Miocene the Andes must have already been high with further uplift at a later stage.

• The writing style at places can be somewhat convolute and could do with rephrasing. A suggestion for the opening sentence would be: "The progressive closure of the Central American Seaway (CAS) and the uplift of the northern Andes profoundly reorganized early Pliocene ocean and atmospheric circulation in the Eastern Equatorial Pacific (EEP)."

Clim. Past Discuss.,
https://doi.org/10.5194/cp-2017-129-AC1, 2018

[Figure]

HOORN Summary The purpose of the paper is to establish the direction of shift of the ITCZ following the closure of the Central American Seaway (CAS) and uplift of the northern Andes. The paper comprises a palynological study of sediments from the interval between 4.7 and 4.2 Ma of the appropriately situated ODP core 1239A. The specific aims are to reconstruct vegetation, climate and topography in this region throughout this time interval. The conclusion is that an (already) high Andean landscape existed at the time, and that both vegetation and landscape during this interval match with a scenario corresponding to a southward shift of the ITCZ. Fluctuations of the ENSO are also considered. The results are in accordance with other paleoceanographic data in the region.

Main comments: There is a shortage of continuous records from the Pliocene in the eastern Pacific that reflect hydrological and climatic change in the region. This paper aims to fill this gap. However, the dataset makes it hard to see the big changes that one would expect from the text. If possible the dataset should be extended with additional data to which are referred in the text.

RESPONSE We emphasize in the text that the changes within the analyzed time window are rather small (e.g. line 20 "stable, permanently humid conditions"; line 271 "During the early Pliocene, no profound changes in the vegetation occur"). We prepare an extra figure (Fig 4A in supplementary file) showing both the data discussed in the manuscript and the data of a pilot study to show the long-term trends. It should be kept in mind that the age model for the period between 5 and 6 Ma is based on shipboard data and less detailed. We will adapt the text accordingly and include: "Percentages of humidity indicators hint to slightly drier conditions at the beginning of the Pliocene. A trend towards higher palynomorph concentrations is found for the period from 6 to 2 Ma. Grass pollen percentages remain low indicating mainly closed forest at altitudes below the Páramo. Representation of lowland rainforest was low around 4.7 Ma, increased by 4.5 Ma, declined again to low levels around 3.5 Ma, and rose to remain at higher levels during the Pleistocene. Continuous presence of pollen and spores from the Páramo indicates that the northern Andes had reached high altitudes in Ecuador before the Pliocene."

HOORN The interaction of Andean uplift, closure of the CAS, shifting ITCZ and ENSO altogether make it quite a daunting task to interpret the palynological diagram an assign changes to specific causes. The case is clearly made and looked at from all angles. Question: Is there a chance that some of the subtle changes in the diagram can be related to the Pliocene uplift pulses in the Andes and related atmospheric changes? Such pulses are postulated in tectonic reconstructions (e.g. Anderson et al., 2015, Geosphere) and are mentioned by authors in the paragraph starting at line 464.

RESPONSE We do not understand this question. We cannot find a paper of Anderson et al. in Geosphere, vol. 11 (2015). We therefore assume that meant is the paper of Anderson et al. in GSA Bulletin that we cite on line 422. However, this paper does not discuss pulses of uplift but that the uplift since 7.6 Ma was more gradual than hypothesized by, for instance, Hooghiemstra et al. (2006).

HOORN The new dataset further confirm that a high topography (Anderson et al., 2015) and paramo (Bermudez et al., 2015 in Basin Research; Hoorn et al., 2017 in Global & Planetary Change) was in place at least since the early Pliocene. It might be worthwhile highlighting the regional character of this condition?

RESPONSE To highlight the regional character, we'll specify Ecuadorean Andes at line 425: "that the Ecuadorean Andes must have already reached close to modern elevations by the early Pliocene": "in line with inferences of Hoorn et al. (2017) and Bermúdez et al. (2015)." (line 425-6). We'll add the reference of Berúmdez et al. to the list. At line 420 we'll add: "Moreover, phases of major uplift might have strongly differed regionally." We'll also specify Ecuadorean Andes at lines 22 and 357.

HOORN Note that modern type precipitation patterns are likely to have been in place already from middle Miocene onwards (see Kaandorp et al., 2006; Hoorn et al., 2010; Barnes et al., 2012) and this would have required a significant orographic barrier. A high Andes might go as far back as the mid-Miocene, however, first evidence for a paramo is now set as latest Miocene to early Pliocene. Lines 406-407 could be reconsidered in this context.

RESPONSE We'll insert "which probably were more or less in place (Kaandorp et al., 2006; Hoorn et al., 2010)," in line 407 after "Possibly these oceanic reorganizations did not directly trigger modifications of the atmospheric circulation," and add the extra reference to the list.

HOORN The elemental concentrations analysis needs to be better introduced and is currently rather hidden and makes a surprise first appearance in the methods section. In methods also explain why this is a useful additional technique. Part of the text in section 4.3 (line 360 onwards) could be moved to the introduction to explain approach.

RESPONSE We are reluctant to shift section 4.3 to the Introduction as it discusses the interpretation of the elemental ratio results. Instead, we'll shift the section to the beginning of the discussion (new section 4.1 see at next point). To satisfy the valid objection, we'll mention the use of elemental ratios at the end of the Introduction (line 75) inserting: "We also use elemental ratios to estimate variations in fluvial terrestrial input (Ríncon-Martínez et al. 2010)."

HOORN The discussion of the Holocene samples in relation to the Pliocene seems a bit ambivalent and does not form a very good guideline to better understand the new results.

RESPONSE To better explain what guidelines we use to interpret the Pliocene record, we'll make the following changes. We change Fig. 2 (supplementary file) and add to Section 1.1.2 two paragraphs replacing lines 139-143:

"Ríncon-Martínez et al. (2010) showed that the terrigenous sediment supply at ODP Site 1239 during Pleistocene interglacials is mainly fluvial and input of terrestrial material drop to low amounts during the drier glacial stages. Also transport of pollen and spores to the ocean is mainly fluvial (González et al., 2010). High rates of orographic precipitation characterize the western part of equatorial South America. These heavy rains quickly wash out any pollen that might be in the air and result in large discharge by the Ecuadoran Rivers (Fig. 2). Esmeraldas and Santiago Rivers mainly drain the northern coastal plain of Ecuador, and the southern coastal plain is drained by several smaller rivers, which end in the Guayas River. Moreover, the predominantly westerly winds (Fig. 2) are not favorable for eolian pollen dispersal to the ocean. Nevertheless, some transport by SE trade winds is possible and should be taken into account."

"After reaching the ocean pollen and spores might pass the Peru-Chile Trench, which is quite narrow along the Carnegie Ridge, by means of nepheloid layers at subsurface depths. Some northward transport from the Bay of Guayaquil by the Coastal Current (Fig. 2) is likely. However, the Peru-Chile Current flows too far from the coast to have strong influence on pollen and spore dispersal. We consider western Ecuador, northernmost Peru and southwestern Colombia the main source areas of pollen and spores in sediments of ODP Site 1239."

We'll switch sections 3.1 and 3.2 to emphasize the function of the Holocene analysis as a tool for interpretation. We add values for Podocarpus to Figure 3 (see supplementary file). We'll replace "whereof the... Alnus" (line 197) with "During the Holocene Podocarpus is replaced by Alnus as the most abundant upper montane forest tree, although Podocarpus was still abundant during the glacial (González et al. 2010)."

We'll rewrite section 4.1 and put it behind the discussion about the elemental ratios consequently becoming section 4.2. We'll open the rewritten section as follows: "In order to better understand the source areas and transport ways of pollen grains to the sediments, we make a comparison of the results of our two Holocene samples with that of another pollen record retrieved from the Carnegie Ridge southeast of Site 1239 (Figure 2) reflecting rainfall and humidity variation of the late Pleistocene (González et al. 2006). Holocene samples of Site 1239 gave similar results showing extensive open vegetation (indicated by pollen of Poaceae, Cyperaceae, Asteraceae) and maximum relative abundance of fern spores although concentration is low (González et al., 2006). As also indicated by the elemental ratios, fluvial transport of pollen predominates in this area (González et al., 2006; Ríncon-Martínez, 2013). This is understandable as both ocean currents and wind field do not favor transport from Ecuador to Site 1239 (Figure 2)."

HOORN Lines from 313: A rather crucial line comes up here and reads as follows: "unpublished data from the earliest Pliocene show that the percentage of lowland rainforest before 4.7 Ma was very low". The evidence that is presented seems rather subtle and perhaps not iconic for an important vegetation & climate change. The authors allude to data of the earliest Pliocene, which they say strengthen their case. However, they are not visible. If these data belong to the authors it might be timely to include them here (or a selection of them) and make a more compelling case.

RESPONSE We'll introduce a new figure (Fig 4A in supplementary file) with the selected results of the pilot study (see also response above) illustrating the low rainforest pollen percentages prior to 4.7 Ma and, more importantly, the continuously higher values after 3 Ma. We'll correct and precise the description accordingly.

HOORN A map with the scenarios for the changing ITCZ would be welcome. Instead this could also be added to figure 1.

RESPONSE We'll change Figure 2 (which we assume is referred to) to show the windfield together with the resulting precipitation during boreal summer (July), because this is the rainy season in the region. This should illustrate the correspondence of the summer rains in northern South America with the present northern limit of the ITCZ. Furthermore, we'll include a panel with summer SST combined with main ocean currents. The adapted figure is shown separately in the supplementary file. This, together with the revision of Section 1.1.2, should also illustrate the ineffectiveness of transport mechanisms other than fluvial discharge by the Guayas and Esmeraldas Rivers.

HOORN Minor comments: In line 465 Hoorn et al. 2010 are listed as backing up a rapid rise of the region since 4–6 Ma, However we suggest in the mid-Miocene the Andes must have already been high with further uplift at a later stage.

RESPONSE Sorry about that. We'll delete Hoorn et al. (2010) from the list and add it later in the paragraph in the altered sentence: "Possibly these oceanic reorganizations did not directly trigger modifications of the atmospheric circulation, which probably were more or less in place (Kaandorp et al., 2006; Hoorn et al., 2010), but critical periods of uplift influencing atmospheric circulation might have occurred earlier." (see also above).

HOORN The writing style at places can be somewhat convolute and could do with rephrasing. A suggestion for the opening sentence would be: "The progressive closure of the Central American Seaway (CAS) and the uplift of the northern Andes profoundly reorganized early Pliocene ocean and atmospheric circulation in the Eastern Equatorial Pacific (EEP)."

RESPONSE We'll do our best and adopt your suggestion for the opening sentence.

ADDITIONAL REFERENCES

Bermúdez, M.A., Hoorn, C., Bernet, M., Carillo, E., Van der Beek, P.A., Garver, J.I., Mora, J.L., and Mehrkian, K.: The detrital record of late-Miocene to Pliocene surface uplift and exhumation of the Venezuelan Andes in theMaracaibo and Barinas foreland basins, Basin Research, 29, Supplement 1, 370-395, 2017.

Kaandorp, R.J.G., Wesselingh, F.O., and Vonhof, H.B.: Ecological implications from geochemical records of Miocene Western Amazonian bivalves, Journal of South American Earth Sciences, 21, 54-74, 2006.

Clim. Past Discuss.,
https://doi.org/10.5194/cp-2017-129-RC2, 2018

[Figure]

This study generated new vegetation and climate record between 4.7 and 4.2 Ma by pollen analysis of 46 samples from ODP Site 1239A, which is located in the East Equatorial Pacific, a place suitable for investigating the precipitation-related fluvial runoff changes in northwestern South America, thus good for monitoring the past movement of the ITCZ. A major aim of this study is to clarify a mismatch about the ITCZ shift in the early Pliocene between the proxy records and the model simulation, that most proxy data supports a southward shift whereas numerical modelling suggests a northward shift in response to Central American Seaway closure and Andean uplift. Generally, this study fills the blank of well dated hydrological record of the early Pliocene in this region by pollen and spores studies from marine sediment.

Generally, I agree with the comments posted by the other three referees and won't repeat it. Here are some minor suggestions, which I think should help the readers to better understand this research if considered.

Age model. How did the authors establish the age model for the study interval of Site 1239? From Tiedemann et al. (2007)? Why not add the benthic d18O record to the figures and sign labels of Marine Isotope Stages? You cannot just cite a reference to get all the necessary things done.

Continuity of the record. Since other palynological studies of the region have been conducted for the mid-Pliocene to the Holocene, why not combine those records with the new record of the early Pliocene? Are they from the same marine core? The new record depends on 46 samples to cover the time interval of 4.7-4.2 Ma, with an average time resolution of 11 Kyr. In such a relatively short period and with a relatively low time resolution, the authors still recognize four major steps of the vegetation changes, and claim that all the vegetation belts as explained in Figures 3 and showed in Figure 4 display synchronous increase/decrease for each stage. If carefully examining figure 4, the features of the variability of the vegetation belts just constrainedly match those described in the text. The referee RC1 also pointed it out. Increasing the time resolution such as doubling, and filling the hiatus between cores 35X and 36X of Hole 1239A (there should be also vegetation change in this interval), something very different probably could happen. Also as indicated by Referee RC1, the unpublished data which is so important to support the author's conclusion of a low percentage of lowland rainforest before 4.7 Ma should be put together with the presented record of this manuscript. I believe that all readers with interests for the ITCZ shift in the early Pliocene would like to see a continuous record since the early Pliocene rather than a segmented record in such a narrow period.

About permanent Elño, closure of Central American Seaway and Andean uplift. My suggestion is weakening the discussion on these comprehensive topics but focusing on its significance in indicating the hydrological changes. The new pollen records are not strong evidences to support the so ambitious conclusions in the present manuscript.

Clim. Past Discuss.,
https://doi.org/10.5194/cp-2017-129-AC4, 2018

[Figure]

TIAN This study generated new vegetation and climate record between 4.7 and 4.2 Ma by pollen analysis of 46 samples from ODP Site 1239A, which is located in the East Equatorial Pacific, a place suitable for investigating the precipitation-related fluvial runoff changes in northwestern South America, thus good for monitoring the past movement of the ITCZ. A major aim of this study is to clarify a mismatch about the ITCZ shift in the early Pliocene between the proxy records and the model simulation, that most proxy data supports a southward shift whereas numerical modelling suggests a northward shift in response to Central American Seaway closure and Andean uplift. Generally, this study fills the blank of well dated hydrological record of the early Pliocene in this region by pollen and spores studies from marine sediment. Generally,

I agree with the comments posted by the other three referees and won't repeat it. Here are some minor suggestions, which I think should help the readers to better understand this research if considered.

Age model. How did the authors establish the age model for the study interval of Site 1239? From Tiedemann et al. (2007)? Why not add the benthic d18O record to the figures and sign labels of Marine Isotope Stages? You cannot just cite a reference to get all the necessary things done.

RESPONSE We'll add the d18OC.wuellerstorfi data of Tiedemann et al. 2007 to Figure 4 (see supplementary file AC2). However, in this part of the Pliocene the fluctuations in the stable oxygen values are small. Also in the stable oxygen data a gap is present around 4.5 Ma, because they have only been measured on sediments of Hole A (same as the pollen). In the results section, we will specify that we used the Tiedemann et al. (2007) age model.

TIAN Continuity of the record. Since other palynological studies of the region have been conducted for the mid-Pliocene to the Holocene, why not combine those records with the new record of the early Pliocene? Are they from the same marine core? The new record depends on 46 samples to cover the time interval of 4.7-4.2 Ma, with an average time resolution of 11 Kyr. In such a relatively short period and with a relatively low time resolution, the authors still recognize four major steps of the vegetation changes, and claim that all the vegetation belts as explained in Figures 3 and showed in Figure 4 display synchronous increase/decrease for each stage. If carefully examining figure 4, the features of the variability of the vegetation belts just constrainedly match those described in the text. The referee RC1 also pointed it out. Increasing the time resolution such as doubling, and filling the hiatus between cores 35X and 36X of Hole 1239A (there should be also vegetation change in this interval), something very different probably could happen. Also as indicated by Referee RC1, the unpublished data which is so important to support the author's conclusion of a low percentage of lowland rainforest before 4.7 Ma should be put together with the presented record of this manuscript. I believe that all readers with interests for the ITCZ shift in the early Pliocene would like to see a continuous record since the early Pliocene rather than a segmented record in such a narrow period.

RESPONSE We think that 11 kyr sample resolution is not too bad for the Early Pliocene. Please, keep in mind that palynological analysis is very time consuming and needs a palynologist well trained in the specifics of the pollen flora under discussion.

We originally put a zone boundary at the coring gap, which in hindsight was unfortunate and, more important, not backed up by the cluster analysis. The CONISS cluster analysis groups samples from below and above the coring gap together suggesting no fundamental trend changes took place during the period in between as also indicated by the XRF-record. We correct this in the new version. Filling the coring gap would take several months of analysis and might not be strictly necessary.

To put the data in a better perspective and to present a more continuous record, we add analyses from the low-resolution pilot study, as also asked for by Carina Hoorn (Figure 4A in supplementary file AC2). We might draw your attention to the long-term development of the lowland forest.

To our knowledge the only published marine pollen diagrams from the region are those of ODP 677 and TR163-38 covering the past 40 and 15ka, respectively (González et al. 2006), and M772-056 covering the past 11ka (Seillès et al., 2016). There is overlap with the top three samples. We wrote on lines 255-257: "A Holocene pollen record from nearby core TR 163-38 has high similarity to the core top samples in its youngest part, showing increased open vegetation (Poaceae, Cyperaceae, Asteraceae), low percentages of Rhizophora, maximum percentages of fern spores, and low pollen and spores concentrations (González et al., 2006)." To which we'll add: "A pollen record closer to the coast - from the Bay of Guayaquil - also indicate relative open vegetation and drier mid- to late Holocene conditions (Seillès et al. 2015) as does the record of ODP Site 677 from the deep basin northwest of Carnegie Ridge (González et al. 2006)."

Detailed analyses for the mid-Pliocene are in progress and are hopefully published (at least submitted) next year. However, this work will focus on the mid-Piacenzian warm Period and on the intensification of the Northern Hemisphere Glaciations. Those are very different themes and beyond the scope of this paper.

TIAN About permanent Elño, closure of Central American Seaway and Andean uplift. My suggestion is weakening the discussion on these comprehensive topics but focusing on its significance in indicating the hydrological changes. The new pollen records are not strong evidences to support the so ambitious conclusions in the present manuscript.

RESPONSE We agree that the changes in the vegetation coupled to changes in hydrology is the core of our paper. We try to look at the hydrological changes from all perspectives as acknowledged by Carina Hoorn. We think that it is important to do so, because so many globally important drivers influence the hydrology of the region. We also are convinced that our Páramo record can be used as an argument that at least the Ecuadoran Andes were already high in the Early Pliocene (see also the responses to Flantua & Hooghiemstra). We'll check the phrasing of the discussion

Clim. Past Discuss.,
https://doi.org/10.5194/cp-2017-129-AC2, 2018

[Figure]
**Figure 2.** Modern climate (boreal summer) and vegetation and core site positions of ODP Sites 677, 846, 851, 1000, 1239, 1241, Trident core TR163-38, and M772-056 mentioned in the text. **A**. Long-term monthly July precipitation in mm/day (CPC) and wind field (NCEP). July is the middle of the rainy season in northern South America, when the ITCZ is at its northern boreal summer position. Salinity estimates for the Caribbean indicate a position of the ITZC further north during the Pliocene. Direction of wind is not favorable for wind transport of pollen and spores to ODP Site 1239. **B**. Long-term monthly July sea surface temperatures (NODC), surface and subsurface currents of the eastern equatorial Pacific (Mix et al. 2003). NECC, North Equatorial Countercurrent; SEC, South Equatorial Current; PCC, Peru-Chile Current (continuation of the Humboldt Current); CC, Coastal Current; EUC, Equatorial Undercurrent; GUC, Gunther Undercurrent. **C**. Contours, bathymetry (ETOPO1), main rivers in Ecuador, and vegetation. Transport of pollen and spores in the ocean over the Peru-Chile Trench, which is very narrow east of the Carnegie Ridge, probably takes place in nepheloid layers. Páramo vegetation is found between 3200 and 4800 m, upper montane Andean forest (UMF) grows between 1000 and 2300 m, sub-Andean lower montane forest (LMF) between 1000 and 2300 m, and lowland forest (LR) below 1000m. The distribution of desert and xeric shrubs in northern Peru, drier broad-leaved forest, flooded grasslands, and mangroves in Ecuador and Colombia is denoted in different colors (see legend, WWF). Source areas of pollen and spores in sediments of ODP Site 1239 are sought in western Ecuador, northwestern Peru, and southwestern Colombia (see text).

[Figure]

**Figure 3.** Comparison of the palynomorph percentages (based on total pollen and spores) of Podocarpaceae and the different vegetation belts between 2 Holocene samples (black) and Pliocene samples between 4.7-4.2 Ma (box-whisker plots).

[Figure]

**Figure 4A (extra figure).** Pliocene and Pleistocene palynomorph percentages (based on the total of pollen and spores) of ODP Hole 1239A for three vegetation belts, humidity indicators, grass pollen and pollen and spore concentration per ml. 95% confidence intervals as grey bars after Maher (1972).

**Additional Results**. Percentages of humidity indicators hint to slightly drier conditions at the beginning of the Pliocene. A trend towards higher palynomorph concentrations is found for the period from 6 to 2 Ma. Grass pollen percentages remain low indicating mainly closed forest at altitudes below the Páramo. Representation of lowland rainforest was low around 4.7 Ma, increased by 4.5 Ma, declined again to low levels around 3.5 Ma, and rose to remain at higher levels during the Pleistocene. Continuous presence of pollen and spores from the Páramo indicates that the northern Andes had reached high altitudes in Ecuador before the Pliocene.

[Figure]

**Figure 4.** Palynomorph percentages of ODP Hole 1239A for the four vegetation belts and other groups from 4.7 to 4.2 Ma. Grey shading represents the 95% confidence intervals (after Maher, 1972). Vertical black lines delimit the pollen zones. At the top stable oxygen isotopes of the benthic foraminifer *C. wuellerstorfi* (Tiedemann et al., 2007) of ODP Hole 1239A, marine isotope stages (MIS), and elemental ratios of Fe/K from Holes 1239A and 1239B. Ages are from Tiedemann et al. (2007). A coring gap is present in Hole 1239A between 4.45 and 4.55 Ma.

[Figure]

**Figure 5.** Palynomorph percentages of Páramo indicators and Asteraceae Tubuliflorae (excluding Ambrosia/Xanthium T) of the past 6 Ma indicating the presence of Páramo vegetation at least since the late Miocene. 95% confidence intervals (grey bars) after Maler (1972). Ages after Tiedemann et al. (2007).

Age (ka)

[Figure]

Age (Ma)

Table 1: List of identified pollen and spore taxa in marine ODP Holes 1239A (Pliocene samples) and 1239B (core top samples, taxa in grey occurred only in core top samples) and grouping according to their main ecological affinity (Flantua et al., 2014; Marchant et al., 2002).

| Páramo | Upper montane forest | Lower montane forest | Lowland rainforest | Broad range taxa | Humid indicators |
|---|---|---|---|---|---|
| *Polylepis/ Acaena* | Podocarpaceae | Urticaceae/ Moraceae | *Wettinia* | Poaceae | Cyperaceae |
| *Jamesonia/ Eriosorus* | *Hedyosmum* | *Erythrina* | *Socratea* | Cyperaceae | *Ranunculus* |
| *Huperzia* | *Clethra* | *Alchornea* | Polypodiaceae | Tubuliflorae (Asteraceae) | *Hedyosmum* |
| *Ranunculus* | *Morella* | *Styloceras T* | *Pityrogramma/ Pteris altissima T* | Amaranthaceae | *Ilex* |
| *Draba* | Acanthaceae | Malpighiaceae | | Rosaceae | *Pachira* |
| *Sisyrinchium* | Melastomataceae | Cyatheaceae | | *Ambrosia/ Xanthium* | *Morella* |
| *Cystopteris diaphana T* | *Daphnopsis* | *Vernonia T* | | Ericaceae | Malpighiaceae |
| | *Bocconia* | *Pteris grandifolia T* | | *Artemisia* | Cyatheaceae |
| | *Myrsine* | *Pteris podophylla T* | | *Ilex* | *Selaginella* |
| | *Lophosoria* | *Saccoloma elegans T* | | *Thevetia* | *Pityrogramma/ Pteris altissima T* |
| | *Elaphoglossum* | *Thelypteris* | | *Salacia* | *Hymenophyllum T* |
| | *Hypolepis hostilis T* | *Ctenitis subincisa T* | | Bromeliaceae | *Thelypteris* |
| | *Grammitis* | | | Malvaceae | *Ctenitis subincisa T* |
| | *Dodonaea viscosa* | | | Euphorbiaceae | *Alnus* |
| | *Alnus* | | | *Liliaceae* | *Cystopteris diaphana T* |
| | | | | Lycopodiaceae excl. *Huperzia Selaginella* | |
| | | | | *Hymenophyllum T* | |
| | | | | *Calandrinia* | |

[Figure]

Supplementary figure. Pollen percentage diagram against age (Tiedemann et al., 2007), with total counts, percentages of single taxa and groups, pollen zones, CONISS clusters based on the curves of single pollen taxa. On top two samples from the Holocene. Minor ticks denote 1%, major ticks 2%, unless stated differently. This panel shows pollen and spore taxa from mangrove, lowland rainforest and lower montane forest. Panels on the next page show the pollen percentages for taxa from the 
[revised manuscript text omitted]

---

## Author Response (AR2)

We thank the anonymous referee for his/her comments. We would like to remark that the referee erroneously reviewed the original version of the manuscript instead of the revised version. Therefore, some comments do not apply anymore. Below please find a point by point reply to all comments, with our replies marked as "RESPONSE". The changes made in the original manuscript are highlighted in yellow. A marked-up manuscript version is attached.

REFEREE: Dear Editor,

Grimmer et al. present palynological and geochemical data tracing hydrological changes from western equatorial South America during the early Pliocene. Given the existing uncertainties of Pliocene climate variability, the controversial discussed shift of the ITCZ possibly in response to the CAS closure, and existing discrepancy between paleoceanographic and model data I feel that the presented study makes a valuable and important contribution to the community. Generally, I consider the manuscript well written, the discussion poses interesting questions and it should be considered for publication after minor revisions.

From the replies to the previous and detailed reviews I find that the authors have addressed the majority of raised issues. However, some concerns remain for me related to the age model in line with pervious reviewer suggestions. I find this crucial part of the study is too briefly discussed and I would like to invite the authors to provide a more in-depth description of what was actually done. I think this is crucial aspect of this study as their new insights into i.e. the onset of the ITCZ shift, easterly wind intensification and ENSO variability requires a firm age constrain. Thus, things I would like to know are: how many, if any, of the stated biostratigraphic maker fall into the study time frame (4.2.-4.7 Ma), and can thus provide direct age control?

RESPONSE: The identified pollen and spores were not used as biostratigraphic markers, because the oxygen isotope stratigraphy of the applied age model (Tiedemann et al. 2007) provides more precise age constraints.

REFEREE: Where are the referenced benthic stable isotope derived from? No citation is provided in Line 200 about these stable isotope data thus I am to assume this is data has been measured in relation to this study? If so, why is this data and its underlying methodologically not properly included and discussed?

RESPONSE: The benthic stable isotopes were measured by Tiedemann et al. (2007) as referenced in the following sentence (line 206). To make it more clear, another reference was included in line 205.

REFEREE: Why is there no age model figure provided in the supplements or main text to highlight the visually correlation, the biostratigraphic events included in the time frame and sedimentation rates?

RESPONSE: ==We included a new figure with the age-depth model as supplementary material (Fig. S2).==

REFEREE: How did the authors establish that 5 m hiatus relate to roughly 100 kyr? Is this hiatus caused by bottom current changes, turbidites, core loss etc.? Sedimentation rate patterns before and after the hiatus might provide an insight into this question. If bottom current changes play a role could this cause to a bias to the pollen assemblage preserved?

RESPONSE: The age-depth correlation was established by Tiedemann et al. (2007). We simply used linear interpolation to calculate the age of the gap. In the initial version of the manuscript, the gap was erroneously described as "hiatus", but this has been changed to "coring gap", which implies that there is no general lack in the sediment sequence caused by bottom current changes, turbidites etc., but rather a sediment loss during coring (this is, however, only an assumption because no details about the coring gap are given in the original publication (Tiedemann et al. 2007)).

REFEREE: Minor issues:

Lines: 114, 115, 119, 130-135, 162-167, 172-174, 177 please provide proper citations for the information provided in these lines.

RESPONSE: Lines 114, 115: Citations were included already in the previous revision (new version line 110). Line 119: The citation is given after the following sentence (new version line 116). ==Lines 130-135: The citation "Balslev 1988" was moved from line 124 to line 126 to include lines 124-126. Another citation was added (new version line 129). Lines 162-167: A citation was added in line 159. Lines 172-174: A citation was added in line 171. Line 177: The sentence was deleted because it was redundant.==

REFEREE: Lines 175-178: It seems to me that 150-200 km paleo-distance seems a bit far to call this site a direct recorder of fluvial input.

RESPONSE: Other studies of the same site (Rincón-Martinez et al. 2010) and same region (González et al. 2006) have shown that the terrestrial fluvial signal is recorded in the marine sediments. One sediment core analyzed by González et al. is even located twice as far from the shore (400 km).

REFEREE: The Fe/K is used as a tracer of fluvial input but those seem rather mild throughout the whole investigated interval whereas the pollen-based indicator of humid conditions shows much more high amplitude variability. What can cause the dissimilarities?

RESPONSE: The differences in amplitude between the proxies are small. The changes in vegetation are rather subtle as well, which we mention several times (compare lines 20, 339, 496). Fe/K and the pollen group of humid indicators are both used to infer changes in fluvial runoff, but it must be considered that their amplitudes of change cannot be directly compared. There are other hydrological effects which might play a role (e.g. the indicators of humid conditions may also record higher soil moisture which is not necessarily coupled with higher fluvial runoff).

REFEREE: Since you also discussed the possibility of eaolian transport of pollen (Lines 342ff) how would that relate to river run off changes at the same time?

RESPONSE: This relation is described in section 4.3.2. When pollen transport is only fluvial, high precipitation and runoff would coincide with high pollen concentrations. As this is not the case in pollen zone III where the pollen concentration is high despite less humid conditions, additional eolian transport is considered (compare also Fig. 4).

REFEREE: Lines 179: Isn`t the Guayas River a bit far south to be directly linked to Site 1239?

RESPONSE: ODP Site 1239 was also located further south in the Pliocene (compare line 172). Additionally, it was also shown by Rincón-Martinez et al. 2010 that the fluvial signal of the Guayas River is captured by Site 1239.

REFEREE: Lines 182: Please state the mcd depth interval that was investigated as this is not immediately obviously from just stating the core numbers.

RESPONSE: The mbsf depth interval was already added during the previous review phase (line 186).

REFEREE: Lines 182: the modern analogue sample were taken how?

RESPONSE: The modern analogue samples were taken the same way as the other samples.

REFEREE: Lines 201: Please explain what you mean by "indirectly orbital tuned"?

RESPONSE: An explanation was added in line 205.

REFEREE: Line 206: How sampled? U-channels, full cores or discrete samples? I suggest you change "A" to "This".

RESPONSE: The measurements were done on the split cores with a non-destructive technique as described in lines 209-214 and references therein. We prefer to stick with "A".

REFEREE: Line 202: mcd depth of hiatus?

RESPONSE: The information was added (new version line 207).

REFEREE: Line 209: A question for me also remains whether or not the XRF-data was corrected for dead time and sample geometry effects? What about non-linear matrix effects? Also, you state Ca alongside Fe and K but never pick up on it during the discussion? Why did you not use Ti/Ca for terrigenous influx/marine productivity since insights into marine productivity changes might also hold information on ocean circulation changes in relation to i.e. ENSO?

RESPONSE: The data was corrected for dead time. The information was added in line 210. Sample geometry effects and non-linear matrix effects were not considered because it does not apply to the measurement technique. In the methods section, Ca was mentioned as an example when describing the technique. In order to not confuse the reader, the examples of elements in line 212 ("such as Fe, Ca, and K") were removed. We used Fe/K as a tracer for fluvial input, because our study focusses on terrestrial changes in hydrology. The use of Ti/Ca for terrestrial climate reconstructions is problematic as Ca is sensitive to dilution effects.

REFEREE: Line 334: unpublished data by whom?

RESPONSE: Our own data, which has already been included in the revised version.

[revised manuscript text omitted]

344, 84-87, 2014.